# Morphometric responses of two zooxanthellate octocorals along a water quality gradient in the Cuban northwestern coast

**Néstor Rey-Villiers**[1]*, **Alberto Sánchez**[2], **Patricia González-Díaz**[3], **Lorenzo Álvarez-Filip**[1]*

1 Biodiversity and Reef Conservation Laboratory, Unidad Académica de Sistemas Arrecifales, Instituto de Ciencias del Mar y Limnología, Universidad Nacional Autónoma de México, Puerto Morelos, Quintana Roo, México, 2 Centro Interdisciplinario de Ciencias Marinas del Instituto Politécnico Nacional, La Paz, Baja California Sur, México, 3 Centro de Investigaciones Marinas, Universidad de La Habana, La Habana, Cuba,

* nestor@cmarl.unam.mx, nestorrvilliers@gmail.com (NRV); lorenzo@cmarl.unam.mx (LAF)

**Data Availability Statement:** All relevant data are within the manuscript and its Supporting Information files

## Abstract

Octocoral abundance is increasing on Caribbean reefs, and one of the possible causes is their vertical morphological plasticity that allows them to grow above the substrate to reduce the effect of processes that occur in it (e.g., scour by sediments) as well as adapt to environmental gradients. The aim of this study was to determine the morphometric response of two octocorals species (*Eunicea flexuosa* and *Plexaura kükenthali*) with different life strategies in a water quality gradient. The research was carried out between 2008 and 2016 on eight forereefs of northwest Cuba. Different morphometric indicators were measured in the colonies of both species found within a belt transect (100 x 2 m) randomly located at each site. The lowest means in height, diameter, number of terminal branches/colony, cover index, and least arborescent colonies of *E. flexuosa* were detected at the sites with the greatest anthropogenic pollution. The water quality gradient did not explain the variability of the five morphometric indicators of *P. kükenthali*. However, hydrodynamic stress was the factor that most negatively affected the morphometry of this species. The chronic effect of poor water quality over time resulted in more small sized colonies of *E. flexuosa* at the polluted site, probably due to higher mortality. The size distribution of *P. kükenthali* also showed the same trend but at the sites with greater hydrodynamic stress. These results show that the morphometric response of octocorals along a water quality gradient is species-specific. This study suggests that poor water quality decreases the size and thus availability of habitat provided by octocorals sensitive to that factor (e.g., *E. flexuosa*) while other tolerant species (e.g., *P. kükenthali*) could provide the habitat of several organisms in a scenario of increasing anthropogenic pollution.

**Funding:** The research was funded by Universidad Nacional Autónoma de México through a postdoctoral scholarship (CTIC 0856 and 4953) to NRV. This work was also supported by SIP-IPN: 20170255, 2018664, and 20195113 projects of Instituto Politécnico Nacional at CICIMAR-IPN. The data collection was supported by the Environmental Monitoring of the Coastal Marine Area (MAZCO) technical scientific service of the Institute Marine Sciences of Cuba. The funders had no role in study design, data collection and analysis, decision to publish, or preparation of the manuscript.

**Competing interests:** The authors have declared that no competing interests exist.

## Introduction

Coral reef ecosystems are biodiversity hotspots that support high biological productivity and provide goods and services to society. Octocorals contribute substantially to supporting coral reef biodiversity and are the second most common group of sessile organisms on the reefs of the Caribbean and Indo-Pacific [1, 2]. Octocoral communities build three-dimensional structures that create marine forests that are the habitat of various organisms [3, 4]. Unlike the population losses reported for hermatypic corals (i.e., reef-building corals), octocorals show signs of resilience, and their abundance is increasing in several Caribbean reefs [5–7]. One of the possible reasons is biological traits such as the vertical morphological strategy that allows the colonies to rise above the substrate and avoid scour by sediments, shading by macroalgae and overgrowth by benthic organisms [6].

The zooxanthellate octocorals that inhabit shallow coral reefs have proliferated in oligotrophic waters due to the symbiosis with photosynthetic dinoflagellates that provides them with metabolic advantages in nutrition [8]. Therefore, it can be expected that, similar to hermatypic corals, zooxanthellate octocorals will be affected by changes in water quality in reefs throughout the tropics [9–12]. However, these effects do not occur in a generalized way at the group level, and there seems to be a differentiated response among species [12, 13]. Some research shows that nutrient enrichment does not affect some biological and biochemical parameters in zooxanthellate octocorals, and some species can even grow faster and reach larger sizes due to the contributions of nutrients from land runoff [14–16]. Other studies have found that some octocorals show tolerance to nutrient pollution of anthropogenic origin, while other species are sensitive in terms of density and morphology [12, 13, 17, 18].

The species-specific morphological response of octocorals to water quality degradation may be related to the high morphological plasticity shown by these organisms [19, 20] and/or to the sensitivity/tolerance of the species to water quality degradation. Octocorals are known to have high morphological plasticity to adapt to environmental gradients such as variations in depth, wave exposure, sedimentation, light levels, resource availability, and habitat types on reefs [19–23]. For example, the height of some octocorals increases with increasing depth, higher light levels, moderate wave energy, and low sedimentation in Caribbean reefs [21, 23]. Therefore, it is possible that morphological variations can also be reflected in gradients of organic pollution of anthropogenic origin.

Studies carried out on the coast of Havana, Cuba, have shown that organic pollution by wastewater has generated a water quality gradient [12, 24, 25], causing a specific response in different octocoral species. *Plexaura kükenthali* has a higher average height in reefs polluted by organic matter [18]. However, the size distributions of *Eunicea flexuosa* show higher percentages of colonies in the lower size ranges in polluted reefs [18]. These two closely related species of the Plexauridae family have different life strategies. Specifically, *E. flexuosa* is sensitive to organic pollution, while *P. kükenthali* is more tolerant since the density of the latter did not show significant differences between reefs under the influence of polluted river basins versus reference sites [12, 26]. In addition to the response of these species, a decrease in the richness, diversity of species, density of colonies and changes in the composition of the octocoral communities toward more tolerant species have been reported in this same gradient [12, 26–28].

The water quality gradient on the Havana coastline allows for the expansion of knowledge about the chronic effect of organic pollution over time (i.e., variation between years) on the morphology of sensitive (e.g., *E. flexuosa*) and tolerant (e.g., *P. kükenthali*) octocorals. Although height is one of the most used morphometric indicators to evaluate the vertical size of octocorals, it does not reflect the size in the horizontal direction, nor does it integrate the different axes of morphological variation. Therefore, other morphometric indicators including

the total length of the branches, height, diameter, height/diameter ratio, cover index and surface area are measured to determine octocoral size [3, 29–31]. In this study, we use different morphometric indicators that complement the height measurements, and we evaluate whether they show a differentiated response in *E. flexuosa* and *P. kükenthali* along the water quality gradient to answer three research questions: 1) What is the response of different morphometric indicators of *E. flexuosa* and *P. kükenthali* along the water quality gradient? 2) Is the response of the morphometric indicators of both species related to water quality degradation or is it the result of morphological plasticity? 3) Does the chronic effect of decreased water quality impact the height of both species over 9 years (2008–2016 period)? To answer these questions, we determined five morphometric indicators in *E. flexuosa* and *P. kükenthali* and the height of both species between 2008 and 2016. Furthermore, the five morphometric indicators in both species were measured along the water quality gradient described by Rey-Villiers et al. [12, 25] using microbiological, hydrochemical, physical, and stable nitrogen isotope variables. All these variables were related to the morphometric indicators in both species since they were obtained in the same sampling period. In the next section, we will delve into the water quality gradient described on the Havana coast [12, 25]. The response of morphological traits to water quality degradation can provide information on the resilience of octocorals in coastal reefs in the Caribbean.

## Materials and methods

### Water quality gradient in the study area

The study was carried out on coral reefs in an area of the northwestern region of Cuba. A stretch of ∼70 km in length was covered from Playa Salado (Sa) to Boca de Calderas (Ca), with most of the sites located on the coast of Havana, the city with the highest population density in Cuba [32]. The sampling sites were located on forereefs approximately 120 to 700 m from the coast, at a depth of 10 m (Fig 1). The sampling sites are not in protected areas, and thus no permits are required to access them and conduct morphometric measurements on both species. The climatic characteristics, marine current direction, geomorphology, sediment composition and a detailed description of the sites can be found in Rey-Villiers et al. [12, 25].

There are polluted river basins on the Havana coast, such as the Havana Bay and the Quibú, Almendares and Cojímar rivers, which provide organic matter and nutrients to coastal reefs [12, 24, 25]. The water quality gradient was characterized by Rey-Villiers et al. [12, 25] using 15 microbiological, hydrochemical, physical variables, and stable nitrogen isotopes in octocorals (S1 Table). In this research, we used the data of these variables to determine the influence of the gradient on the morphology of *E. flexuosa* and *P. kükenthali*. In general, those variables showed that the sites Parque Antonio Maceo (PAM), La Puntilla (Pu) and the underwater sewage outfall of 180 Street (DS) had poor water quality because they presented the highest concentrations of total and fecal coliform bacteria, fecal streptococci bacteria, heterotrophic bacteria, ammonia, dissolved inorganic nitrogen, greater $\delta^{15}$N of *E. flexuosa* and *P. kükenthali*, higher bottom-sediment accumulation and lower horizontal visibility in the water column (hereinafter water visibility) (Fig 1 and S1 Table). These three sites were under the influence of Havana Bay (e.g., PAM) and the Almendares (e.g., Pu) and Quibú (e.g., DS) rivers. The Sa and Ca sites had the best water quality and were far from these river basins and from the urban, touristic, and industrial development of Havana (Fig 1 and S1 Table). Continuous discharges from the Quibú River can also reach the reefs in front of the Institute of Oceanology (IO) and Club Habana (CH), and together with the 30 Street (C30) site, they had an intermediate level of pollution [12, 25] (Fig 1 and S1 Table). Therefore, this coastline is characterized by the

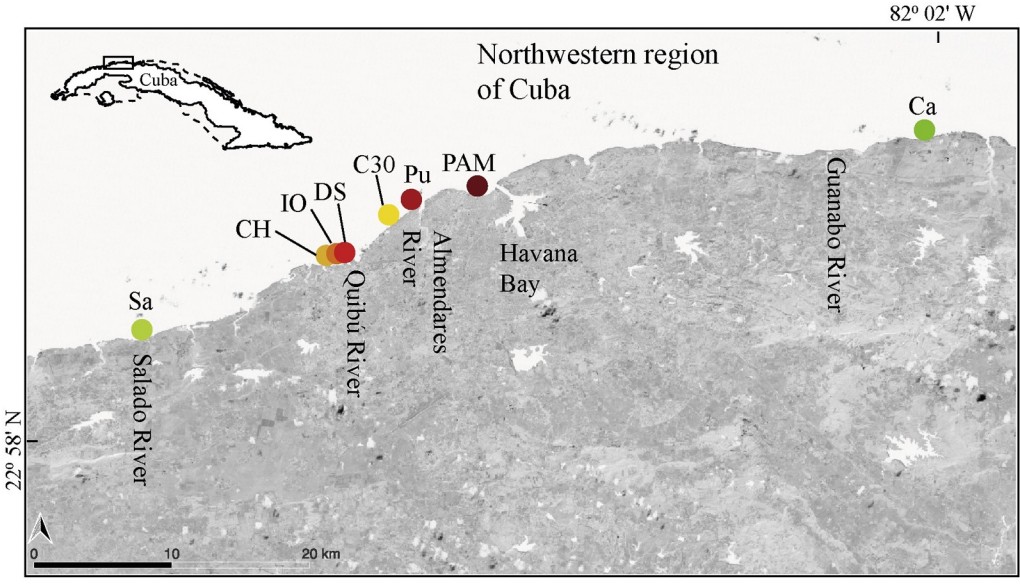

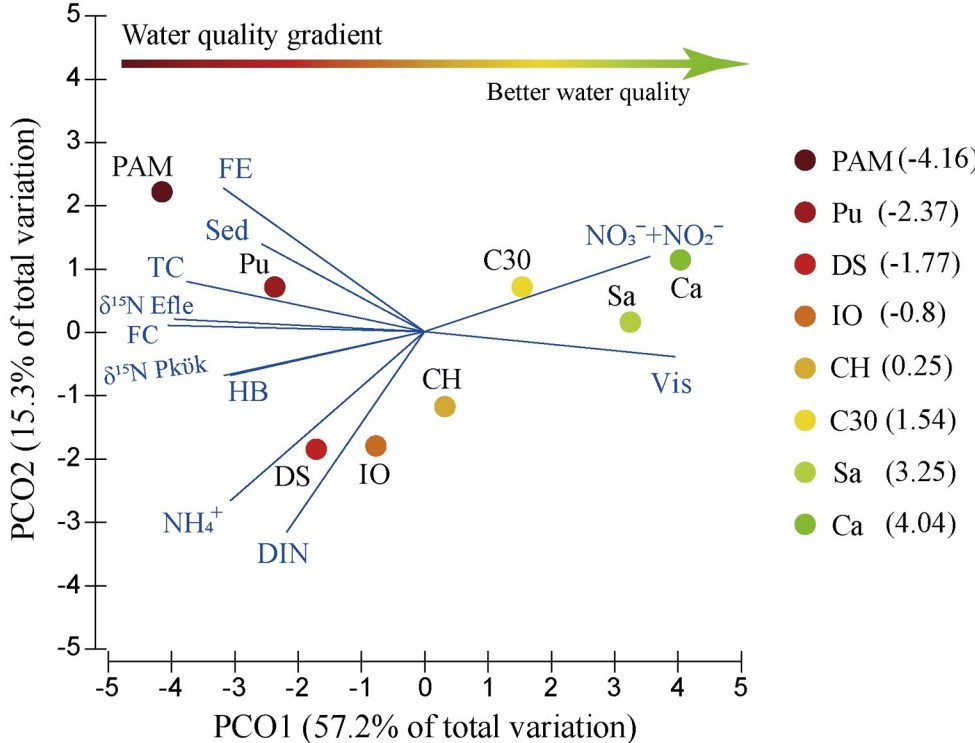

**Fig 1. Map of the sampling sites in an area of northwest Cuba.** Principal coordinates analysis (PCO) plot shows the water quality gradient according to the microbiological, hydrochemical and physical variables and stable nitrogen isotopes of two octocorals. The vectors indicate the strongest Spearman's correlation ($r > 0.7$) between the variables and the PCO1 and PCO2 axes. The number in parentheses is the location of each site along PCO1. Sa: Playa Salado, CH: Club Habana, IO: Institute of Oceanology, DS: underwater sewage outfall of 180 Street, C30: 30 Street, Pu: La Puntilla, PAM: Parque Antonio Maceo and Ca: Boca de Calderas. FC: fecal coliform bacteria, TC: total coliform bacteria, FE: fecal streptococcal bacteria, HB: heterotrophic bacteria, $NH_4^+$: ammonium, $NO_3^- + NO_2^-$: nitrate + nitrite, DIN: dissolved inorganic nitrogen, Vis: water visibility (m), Sed: bottom-sediment accumulation, *Efle*: *E. flexuosa*, *Pkük*: *P. kükenthali*.

presence of a water quality gradient generated by constant discharges from polluted river basins and wastewater discharges from various underwater sewage outfalls [12, 24, 25].

The microbiological and hydrochemical variables and water visibility were determined between 2008 and 2016, the bottom-sediment accumulation was estimated in 2011, and the collections of both octocorals to determine the stable nitrogen isotopes were made in 2016. A detailed description of the methodology used to determine all these variables can be found in Rey-Villiers et al. [12, 25], except for hydrodynamic stress. Hydrodynamic stress, defined as the turbulent movement of water generated by waves, was inferred by monitoring the octocoral communities between 2002 and 2015. For this purpose, the hydrodynamic stress index (HSI) was used. Briefly, this index is a proxy for hydrodynamic stress and is calculated from the sum of the relative abundance of 11 octocoral species considered tolerant to this factor [33].

## Morphometric indicators

At each study site, a belt transect (100 x 2 m) was carried out randomly on the coralline–rocky substrate parallel to the coast at a depth of 10 m. The colonies of *E. flexuosa* and *P. kükenthali* in the transect were measured with a tape measure with 0.1 cm precision. The height from the base to the apex of the colony (H), the height from the first branch to the apex of the colony (h), and the maximum (D) and minimum (d) diameter of the colony in top view were measured (S1 Fig), and the number of terminal branches per colony was counted in 2010. With the morphometric measurements of h, D, and d, a cover index (CvI) was calculated that represents an estimate of the colony volume [29]. This index is calculated from two equations: 1) when the D/d ratio $<2$, $CvI = (D + d/4)^2 \pi h \times 10^{-3}$, and 2) when the D/d ratio $\geq 2$, $CvI = Ddh/2 \times 10^{-3}$. The height of *E. flexuosa* and *P. kükenthali* was weighted to the maximum diameter (H/D ratio) to obtain a comparative measure of the arborescence of the colonies. Therefore, the response of five morphometric indicators (H, D, H/D ratio, CvI and number of terminal branches/colony) in both octocoral species was evaluated along the water quality gradient.

The chronic effect of the water quality gradient on the morphometry of *E. flexuosa* and *P. kükenthali* was evaluated through the height of the colonies at each site and in 2008, 2010 and 2016. These measurements allowed quantification of the number of recruits (i.e., colonies $\leq 5$ cm in H) [34] of both species. The sample size of *E. flexuosa* varied between 93 and 122 colonies per site, and that of *P. kükenthali* varied between 29 and 120 colonies. The cumulative curves of the standard error versus number of colonies for the height of both species showed asymptotic behavior (S2 Fig). This reflected an adequate sample size to compare the morphometry of the two species between sites and years.

## Data analysis

To evaluate the response of the morphometric indicators of *E. flexuosa* and *P. kükenthali* to the water quality gradient, a univariate analysis of variance was performed with 999 unrestricted permutations of the raw data [35]. Euclidean distance was used as a measure of similarity. Pairwise comparisons using PERMANOVA showed that pairs of means had significant differences. The largest effect sizes (difference between two means) were determined for the morphometric indicators of both species. The 95% confidence interval (95% CI) of the mean of the morphometric indicators and of the effect sizes were calculated from the Monte Carlo test with 10,000 iterations and by the percentile method using the program PopTools 3.0.5.

Data normality was verified through the Shapiro–Wilk test using the Statistica 12.0 and SigmaPlot 12.0 programs for regression analysis. To determine if the response of the morphometric indicators of *E. flexuosa* and *P. kükenthali* was related to the water quality degradation or to morphological plasticity, a linear regression analysis was performed between the mean of each

morphometric variable and the values of the location of each site along the PCO1 axis in Fig 1. This axis explained the greatest variation of sites in correspondence with the water quality gradient. Linear regression was also performed between the mean of each morphometric indicator and the hydrodynamic stress and between the height and abundance (colonies/m$^2$) of both octocoral species. Data on the abundance of both species were obtained from Rey-Villiers et al. [12, 25]. Additionally, Pearson's correlation coefficient was calculated between the morphometric indicators of both species and the microbiological, hydrochemical and physical variables and stable nitrogen isotopes. Variables were resampled separately, without replacement, and the null distribution of the data was constructed with the Monte Carlo test with 10,000 iterations. This distribution made it possible to determine the probability that the pairs of variables were randomly correlated. The sampling sites were used as replicates (n = 8). The calculations were performed in Microsoft Office Excel and PopTools 3.0.5.

To evaluate the chronic effect of water quality degradation (years 2008, 2010 and 2016) on the height of *E. flexuosa* and *P. kükenthali*, size distributions and a bifactorial analysis of variance were performed (factors: site, year and site x year) with permutations [35]. The size distribution of both species was compared between years at each site and between sites using permutational analysis of multivariate dispersions (PERMDISP). The Euclidean distance to the spatial median and 999 permutations of the absolute minimum deviation of the residuals were used. In addition, the distributions that had significant differences were detected from the pairwise comparisons made with PERMDISP. The estimates of components of variation (ECV) in the two-factor design allowed quantifying the magnitude of the effects of each factor to explain the variability in the height of both species. The procedure was based on 999 permutations of residuals under a reduced model to generate the distribution of the data. For this, two similarity matrices were constructed with the Euclidean distance. The calculations were made with the PRIMER 6 and PERMANOVA programs.

## Results

### Response of the morphometric indicators of *E. flexuosa* and *P. kükenthali* to the water quality gradient

The mean diameter of *E. flexuosa* in the study area was 17.6 cm (95% CI: 16.7–18.6 cm; range: 1–112 cm). The diameter of this species varied significantly over the water quality gradient (Pseudo-$F_{7, 946}$ = 50.3, P = 0.001), with the highest values in Sa and Ca and the lowest in Pu, PAM and DS (Fig 2). The largest effect size was quantified between Ca and Pu (24.1 cm, 95% CI: 21.1–27.3 cm). The rest of the morphometric indicators of *E. flexuosa* had a similar variation as the colony diameter, which suggests that there is no differential response of these variables to the water quality gradient. The mean number of terminal branches/colony was 46 (95% CI: 36–58 branches, range: 2–640 branches) and showed significant differences over the water quality gradient (Pseudo-$F_{7, 178}$ = 8.3, P = 0.001). The number of terminal branches/colony was higher at Ca and Sa, while it was lower at Pu, PAM and DS (Fig 2). The largest effect size was quantified between Ca and Pu (124 branches, 95% CI: 115–133 branches). The mean cover index was 6.4 dm$^3$ (95% CI: 5.4–7.5 dm$^3$, range: 0.0002–267.7 dm$^3$) and had significant spatial variation (Pseudo-$F_{7, 946}$ = 14.9, P = 0.001). The highest cover indices were recorded at Ca, CH and Sa, and the lowest were recorded at Pu, PAM and DS (Fig 2). The largest effect size was calculated between Ca and Pu (16.3 dm$^3$, 95% CI: 11.2–22.6 dm$^3$). The relationship between the height/diameter of *E. flexuosa* also presented significant differences between sites (Pseudo-$F_{7, 947}$ = 16.8, P = 0.001). The highest value was significantly detected at Pu, and the lowest was detected at Ca and Sa (Fig 2). The largest effect size was quantified between Pu and Ca (1.2, 95% CI: 0.96–1.46).

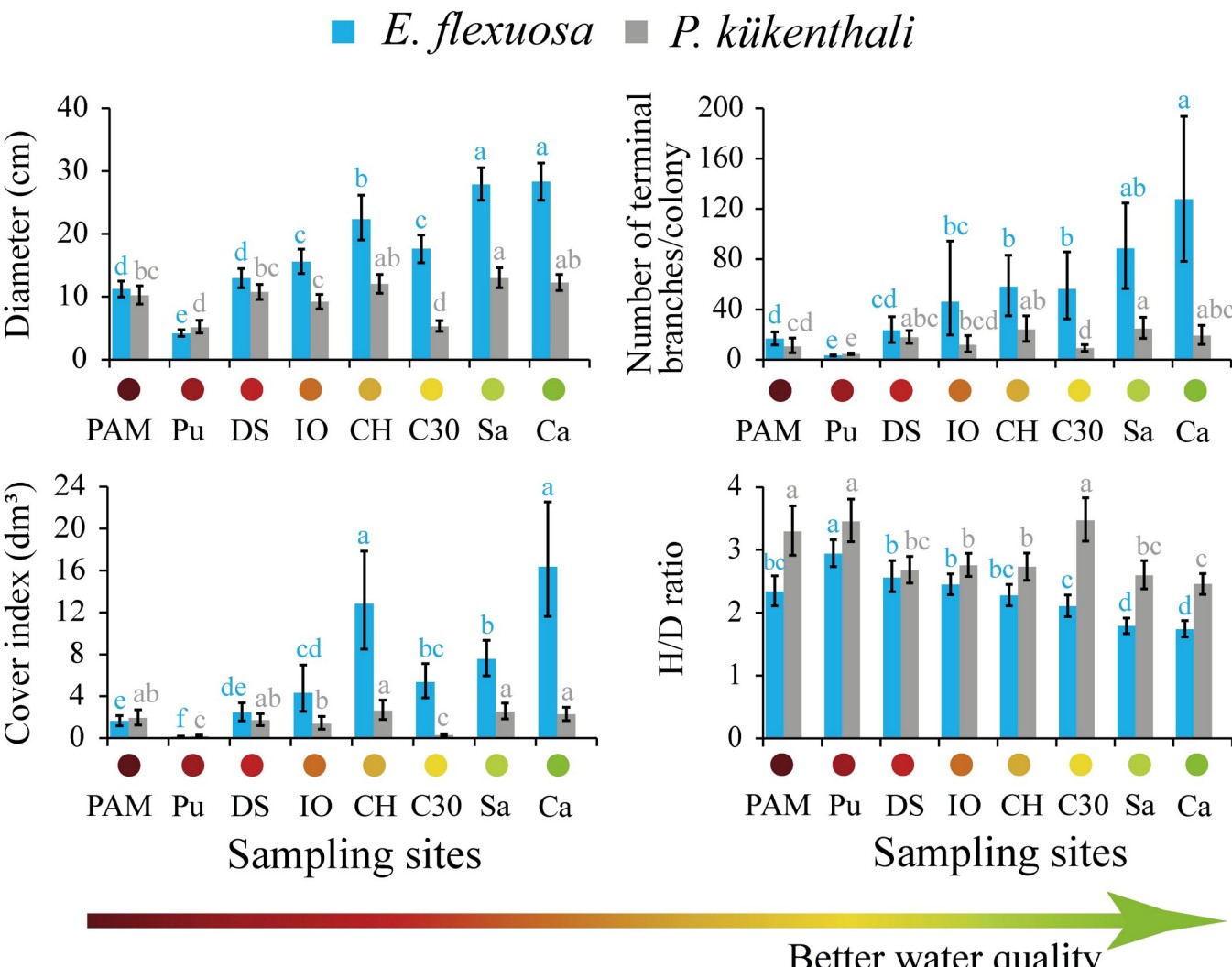

**Fig 2. Morphometric indicators of *E. flexuosa* (blue) and *P. kükenthali* (gray) throughout the water quality gradient.** The mean and 95% CI of the maximum diameter (cm), number of terminal branches/colony, cover index (dm³) and height/diameter ratio (H/D) are shown. Different lowercase letters of the same color indicate significant differences between mean pairs. Sampling sites codes are presented in Fig 1.

The response of the morphometric indicators of *P. kükenthali* to the water quality gradient was different from that observed in *E. flexuosa*. The average diameter of *P. kükenthali* was 10.1 cm (95% CI: 9.6–10.6 cm, range: 1–55.3 cm) in the study area. Although the diameter varied significantly over the water quality gradient (Pseudo-$F_{7, 885}$ = 17.5, P = 0.001), the lowest values were detected only at Pu and C30, and there were no differences between two of the sites with worse (PAM and DS) and better (Ca) water quality (Fig 2). The largest effect size was quantified between Sa and Pu (7.8 cm, 95% CI: 5.7–10 cm). The average number of terminal branches/colony was 12 (95% CI: 10–14 branches, range: 2–88 branches), and there were significant differences over the water quality gradient (Pseudo-$F_{7, 216}$ = 9.9, P = 0.001). The number of terminal branches/colony was significantly higher at Sa, CH and Ca than at Pu and C30, and no differences were detected between two of the sites with worse (PAM and DS) and better (Sa and Ca) water quality (Fig 2). The largest effect size was determined between Sa and Pu (20 branches, 95% CI: 19–21 branches). The cover index was 1.7 dm³ (95% CI: 1.5–2 dm³, range:

0.001–34.9 dm³). This index showed significant spatial variation (Pseudo-$F_{7, 884}$ = 6.6, P = 0.001), with the highest values at CH, Sa and Ca and the lowest at Pu and C30, without significant differences between two of the sites with worse (PAM and DS) and better water quality (Sa and Ca) (Fig 2). The largest effect size was calculated between CH and Pu (2.5 dm³, 95% CI: 1.4–3.8 dm³). The relationship between the height/diameter of *P. kükenthali* showed significant spatial variation (Pseudo-$F_{7, 885}$ = 8.5, P = 0.001) but without detecting differences between some sites with worse (DS) and better water quality (Sa and Ca) (Fig 2). The largest effect sizes were quantified between C30 and Ca (1.95% CI: 0.63–1.42) and between Pu and Ca (1.95% CI: 0.61–1.4).

The water quality gradient significantly explained 61–87% of the variation in the morphometric indicators of *E. flexuosa*, indicating that water quality degradation had a negative influence on the morphometry of this species. However, the response of the morphometric indicators of *P. kükenthali* did not show a significant relationship with the water quality gradient (Fig 3).

The height, diameter, number of terminal branches/colony and cover index of *E. flexuosa* showed a significant negative correlation with fecal coliforms, total coliforms, fecal streptococci, $NH_4^+$ concentration, bottom-sediment accumulation, and $\delta^{15}N$ in this species (S2 Table). The height, diameter, number of terminal branches/colony and cover index of *E. flexuosa* had a significant positive correlation with the water visibility and $NO_3^- + NO_2^-$ concentration (except for cover index with $NO_3^- + NO_2^-$) (S2 Table). The number of terminal branches/colony had a significant negative correlation with heterotrophic bacteria. In *E. flexuosa*, the height/diameter ratio had a significant positive correlation with fecal coliform bacteria, total coliforms, $NH_4^+$ concentration, DIN, and $\delta^{15}N$ (S2 Table). This morphometric indicator showed a significant negative correlation with the $NO_3^- + NO_2^-$ concentration and water visibility (S2 Table). Conversely, the responses of all the morphometric indicators of *P. kükenthali* were significantly related to hydrodynamic stress, a factor that did not explain the variability in the morphometric indicators of *E. flexuosa* over the water quality gradient (Fig 4 and S2 and S3 Tables).

### Chronic effect of water quality degradation on the height of *E. flexuosa* and *P. kükenthali*

The size frequency distributions of *E. flexuosa* and *P. kükenthali* did not show significant differences between years at any of the sampling sites (S3 and S4 Figs). Therefore, the colonies were grouped to compare the size frequency distribution between sites over time (2008–2016 period). The size frequency distribution of *E. flexuosa* showed significant differences along the water quality gradient (F = 68.6, P = 0.001), with the highest dispersions at Sa, Ca, IO and CH (Fig 5). The chronic effect of water quality degradation over time on the height of *E. flexuosa* was a higher percentage of colonies in the smaller size ranges at the Pu, PAM and DS sites (Fig 5). Furthermore, at these sites, most of the colonies did not exceed 40 cm in height. For example, at Pu, no colony exceeded 40 cm in height; at DS, only 5 colonies reached the range of 40–50 cm in height; and at PAM, we only detected four colonies between 40–50 cm and two colonies between 50–60 cm and 60–70 cm tall (Fig 5).

The size frequency distribution of *P. kükenthali* also had significant differences between sites (F = 10.6, P = 0.001), with the highest dispersions at Sa, Ca, CH and IO and the lowest dispersions at Pu and C30. At these last two sites, there was a higher percentage of colonies in the smaller size ranges, and no colony reached more than 40 cm in height. Apparently, there was no clear chronic effect of water quality degradation on the height of this species since the size distribution did not show significant differences between one of the sites with worse (DS) and better (Sa and Ca) water quality (Fig 6).

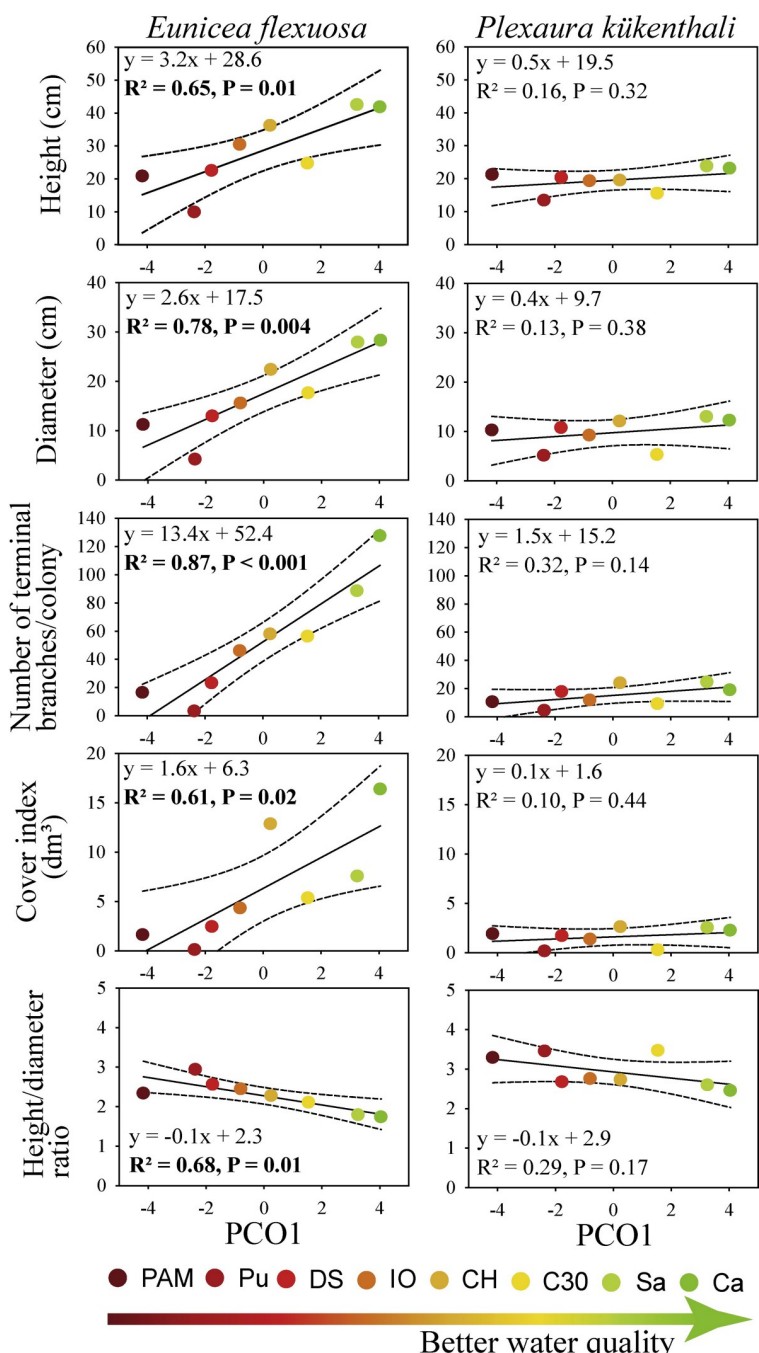

**Fig 3. Linear regression between the water quality gradient explained through the PCO1 of Fig 1 and the morphometric indicators of *E. flexuosa* and *P. kükenthali*.** The dotted lines represent the upper and lower 95% confidence intervals.

The mean height of *E. flexuosa* including all the sampling sites and years (2008, 2010 and 2016) was 29 cm (range: 2–100.6 cm) and that of *P. kükenthali* was 20.2 cm (range: 2–60.5 cm). The chronic effect of the water quality gradient on the height of *E. flexuosa* was up to 64 times greater than the variations between years and up to 35 times greater than the specific temporal variation at the sites (Table 1). This evidence suggests that the height of *E. flexuosa*

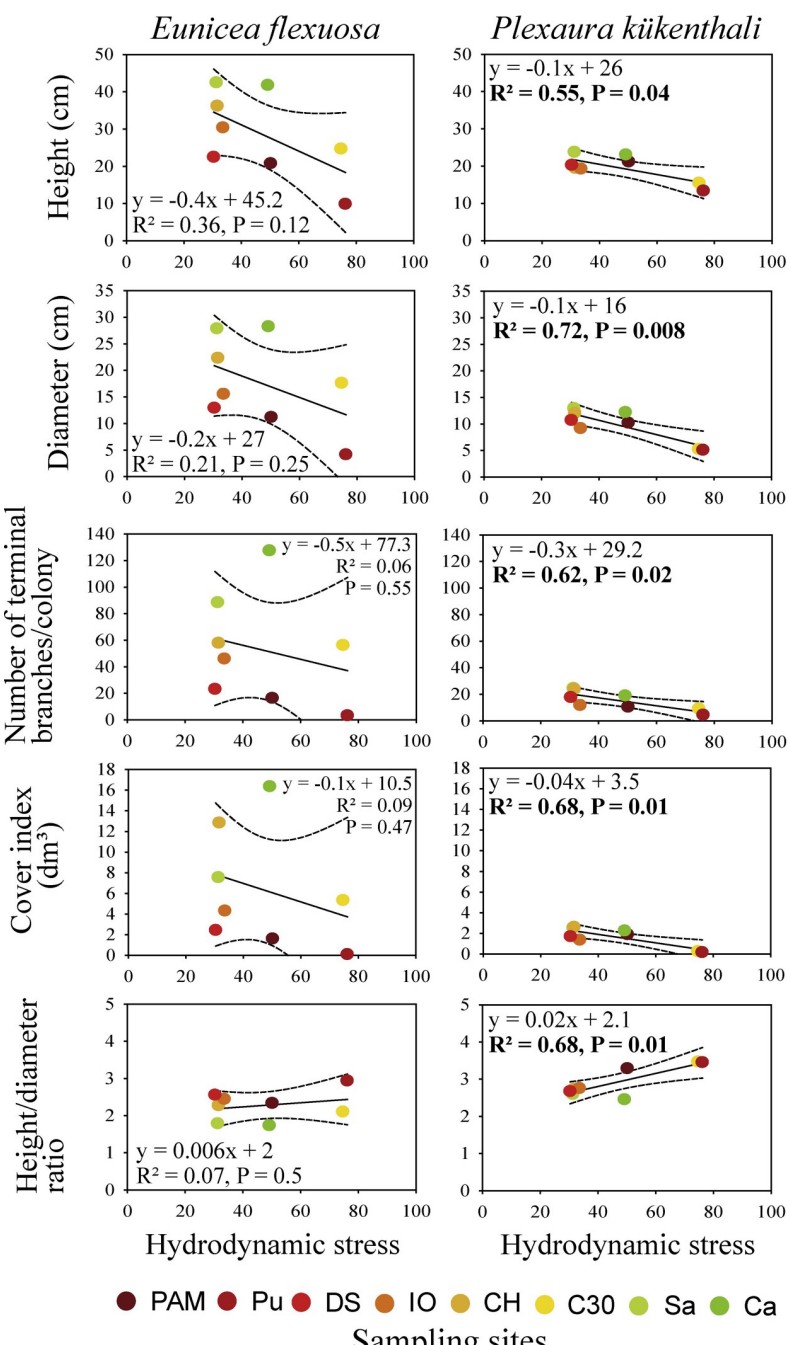

**Fig 4. Linear regression between the morphometric indicators of *E. flexuosa* and *P. kükenthali* and hydrodynamic stress.** The dotted lines represent the upper and lower 95% confidence intervals.

varies depending on the water quality at the sites. In fact, the mean height of *E. flexuosa* was significantly lower at the sites with degraded water quality (Pu, PAM, DS, IO) than at the sites with better water quality (Sa and Ca) over time (Fig 7). In contrast, the water quality gradient and the temporal variation explained a similar percentage of the variability in the height of *P. kükenthali*. The effect of the water quality gradient on the height of *P. kükenthali* had a lower magnitude than that on the height of *E. flexuosa* (Table 1). These results suggest that the height

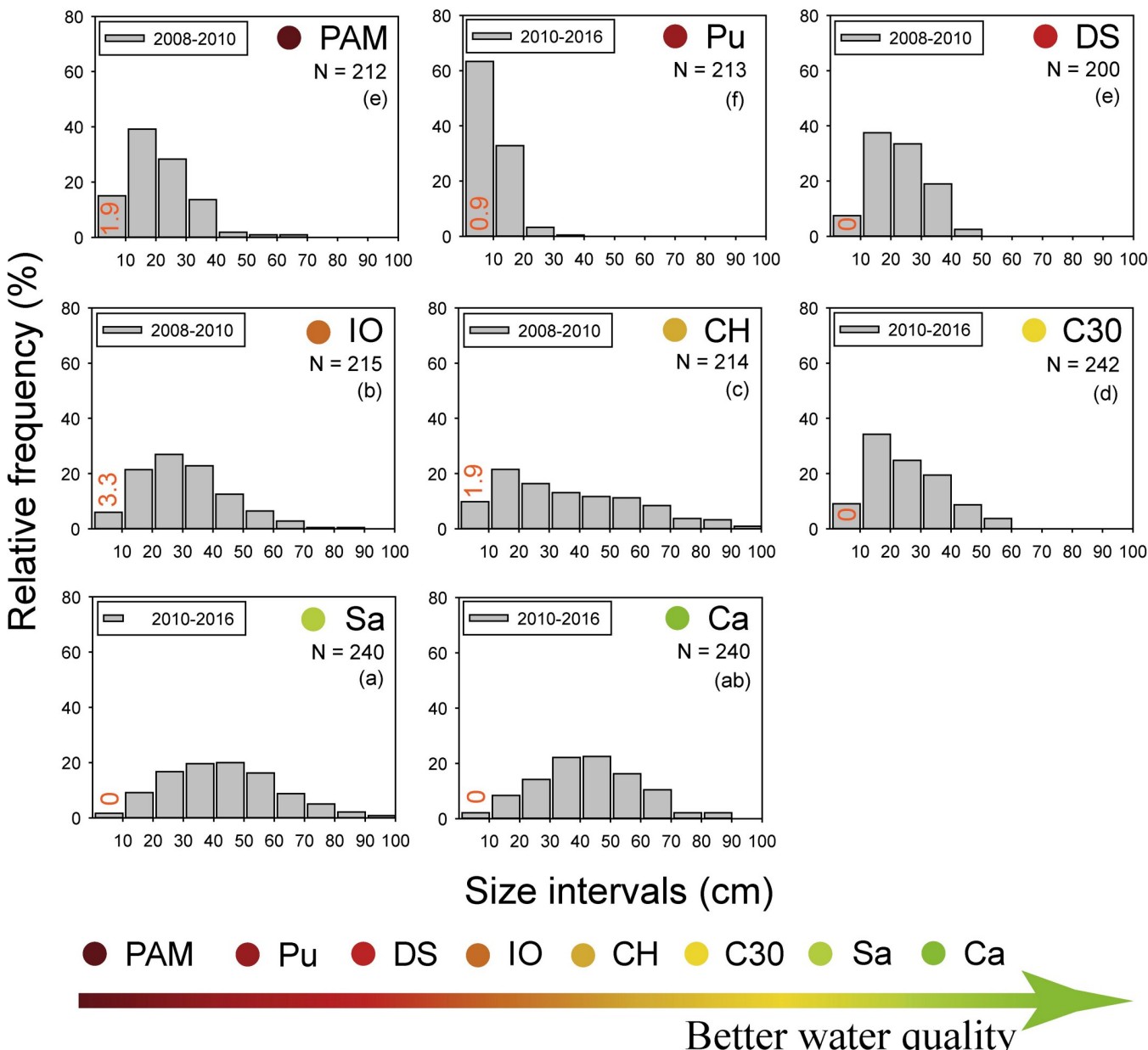

**Fig 5. Relative size-frequency (based on height) distributions of *E. flexuosa* along the water quality gradient over the 2008–2016 period.** The percentage of recruits is shown in the ≤ 10 cm size interval (orange numbers). The results of pairwise comparisons of the PERMDISP test are shown (lowercase letters in brackets). N: sampling size. Sampling sites codes are presented in Fig 1.

of both species shows a specific response to the water quality over time. In fact, the height of *P. kükenthali* varied over time regardless of the water quality at the sites, with differences between sampling years both at the sites with worse (DS) and better (Sa and Ca) water quality (Fig 7). For example, at DS, the mean height increased significantly in 2010 and did not show significant differences from the sites with better water quality (Sa and Ca). The mean height of *P. kükenthali* was only significantly lower over time at Pu and C30 compared to the rest of the sites (Fig 7). These results do not show clear evidence of the chronic effect of water quality on the height of *P. kükenthali*.

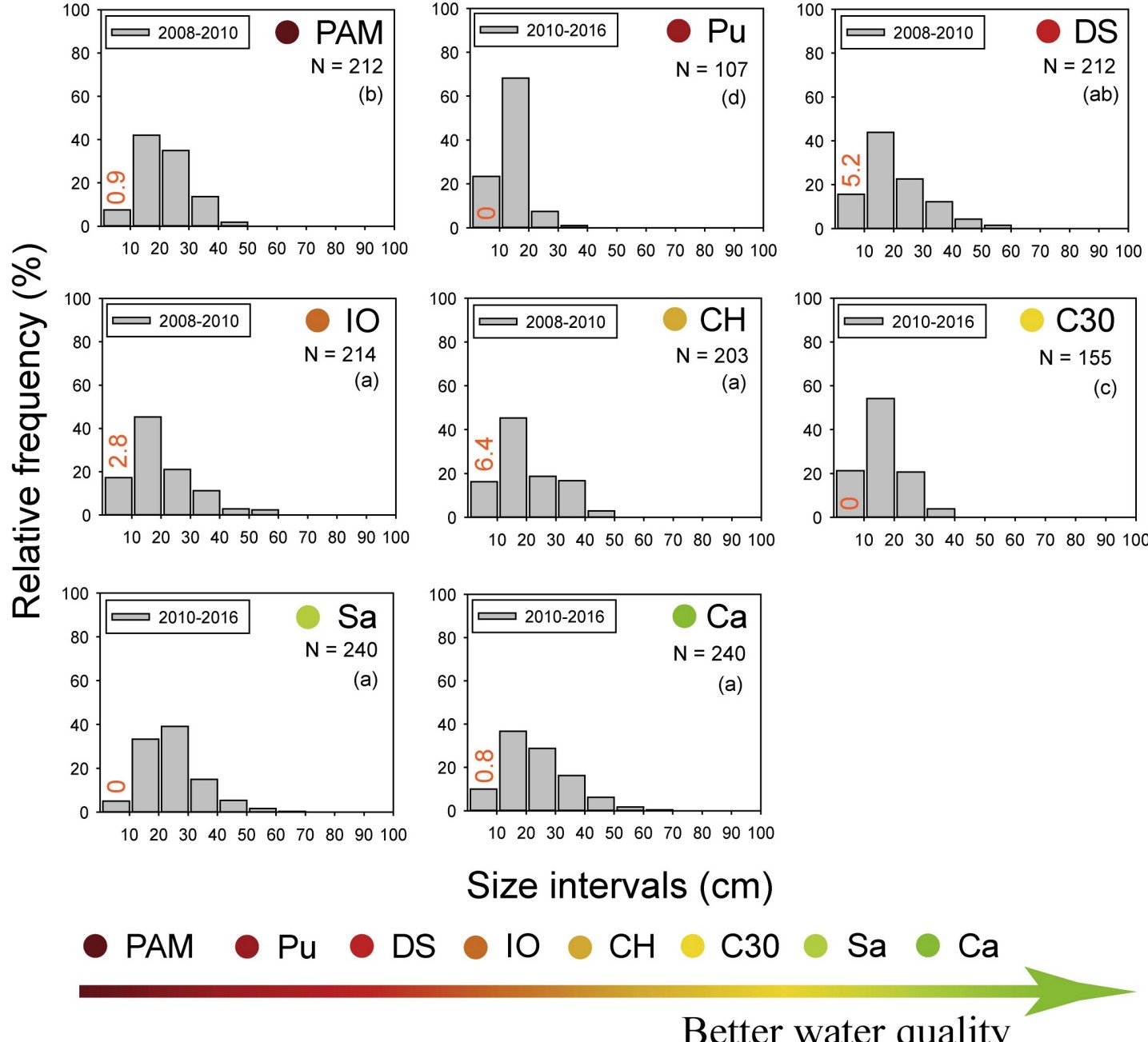

**Fig 6. Relative size-frequency (based on height) distributions of *P. kükenthali* along the water quality gradient over the 2008–2016 period.** The percentage of recruits is shown in the ≤ 10 cm size interval (orange numbers). The results of pairwise comparisons of the PERMDISP test are shown (lowercase letters in brackets). N: sampling size. Sampling sites codes are presented in Fig 1.

## Discussion

### Response of the morphometric indicators of *E. flexuosa* and *P. kükenthali* to the water quality gradient

The smaller diameter, number of terminal branches/colony, and cover index of *E. flexuosa* at the sites with the highest organic pollution indicate that the discharges from Havana Bay and

**Table 1. Magnitude of the effects of factors on the height of *E. flexuosa* and *P. kükenthali*.** The effect of each factor was evaluated by the estimates of components of variation (ECV) using a PERMANOVA test.

| Variable | Factor | ECV | ECV (%) |
|---|---|---|---|
| Height of *E. flexuosa* | Site | 134.4 | 38.2 |
| | Year | 2.2 | 0.6 |
| | Site x year | 3.8 | 1.1 |
| | Residual | 211.5 | 60.1 |
| Height of *P. kükenthali* | Site | 10.1 | 8.9 |
| | Year | 8.1 | 7.1 |
| | Site x year | 5.6 | 4.9 |
| | Residual | 89.7 | 78.9 |

the Almendares and Quibú rivers negatively affect the morphometry of this species. The larger effect sizes of these variables between highly polluted sites and sites distant from urban, touristic, and industrial development in Havana highlight the negative influence of these river basin discharges on the *E. flexuosa* morphometric indicators. In this same gradient, the greater height/diameter ratios reflect fewer arborescent colonies of *E. flexuosa* on the reefs with greater pollution. In fact, the water quality gradient significantly and similarly explained the decrease in the five morphometric indicators, suggesting that there is no differential response among the indicators. In addition, the significant correlations between the five morphometric indicators of *E. flexuosa* and the microbiological variables, $NH_4^+$ concentration, and $\delta^{15}N$ in the tissue of this species support that the decline in water quality negatively affects its morphometry. Additionally, the bottom-sediment accumulation and the low water visibility also negatively influenced the morphometry of this species. These findings are similar to the results of Tsounis et al. [23], who detected greater heights of *E. flexuosa*, *Antillogorgia americana* and *Gorgonia ventalina* in a reef with less sedimentation and greater light availability in the U.S. Virgin Islands, Caribbean Sea. In addition, the species *E. flexuosa* and *Plexaura homomalla* were less arborescent where there was less light availability in the Panamanian Caribbean [36], which coincided with the less arborescent colonies of *E. flexuosa* at the sites with lower visibility in the water quality gradient on the coast of Havana.

In contrast, the water quality gradient did not significantly explain the morphometric variability of *P. kükenthali*. This suggests that there is a differentiated response in the morphometry of both species to this factor, which has been observed in other cnidarians, such as in the diameter of hermatypic corals [37, 38]. The differentiated morphometric response of the two octocoral species to the water quality gradient may be related to their high morphological plasticity [19, 20] or to their life strategies. *E. flexuosa* has been considered sensitive to organic pollution, nutrient enrichment, bottom-sediment accumulation, and low water visibility, while *P. kükenthali* is more tolerant of organic pollution [12]. This is attributed to the significantly lower density of *E. flexuosa* colonies (colonies/m$^2$) at sites influenced by polluted river basin discharges compared to reference sites. Furthermore, the density of *E. flexuosa* showed a significant negative correlation with several microbiological and physical variables [12]. As for *P. kükenthali*, its density increased at sites affected by polluted river basin discharges and did not show significant differences between the reefs near Havana Bay (PAM) and the reference sites (Sa and Ca) [12]. Therefore, the sensitivity of *E. flexuosa* and the tolerance of *P. kükenthali* to water quality degradation seems to be one of the life strategies that influences the differentiated morphometric response of both species.

It is notable that despite the sensitivity of *E. flexuosa* to organic pollution, this species has shown high morphological plasticity to other environmental gradients, such as depth and

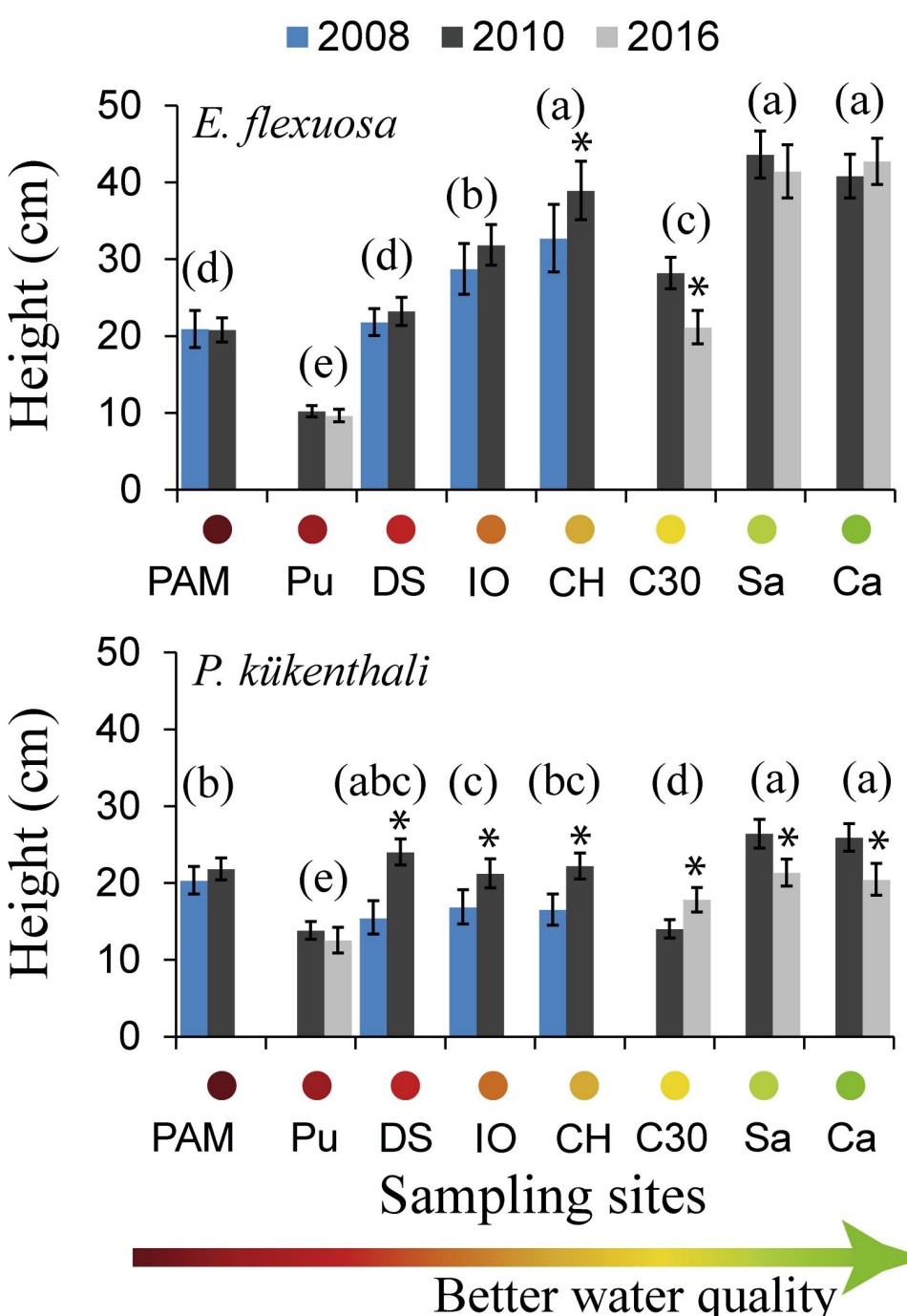

**Fig 7.** Height (cm) of *E. flexuosa* (top) and *P. kükenthali* (bottom) throughout the water quality gradient over sampling years. The mean and 95% CI are shown. Different lowercase letters indicate significant differences between sites, and asterisks indicate significant differences between sampling years. Sampling sites codes are shown in Fig 1.

habitat types in reefs [19, 20]. However, the negative effect of organic pollution on *E. flexuosa* morphometry does not seem to be related to morphological plasticity, due to the decrease in abundance of this specie. The decrease in *E. flexuosa* height significantly explained the reduction in its abundance along the water quality gradient between 2008 and 2016 (Fig 8).

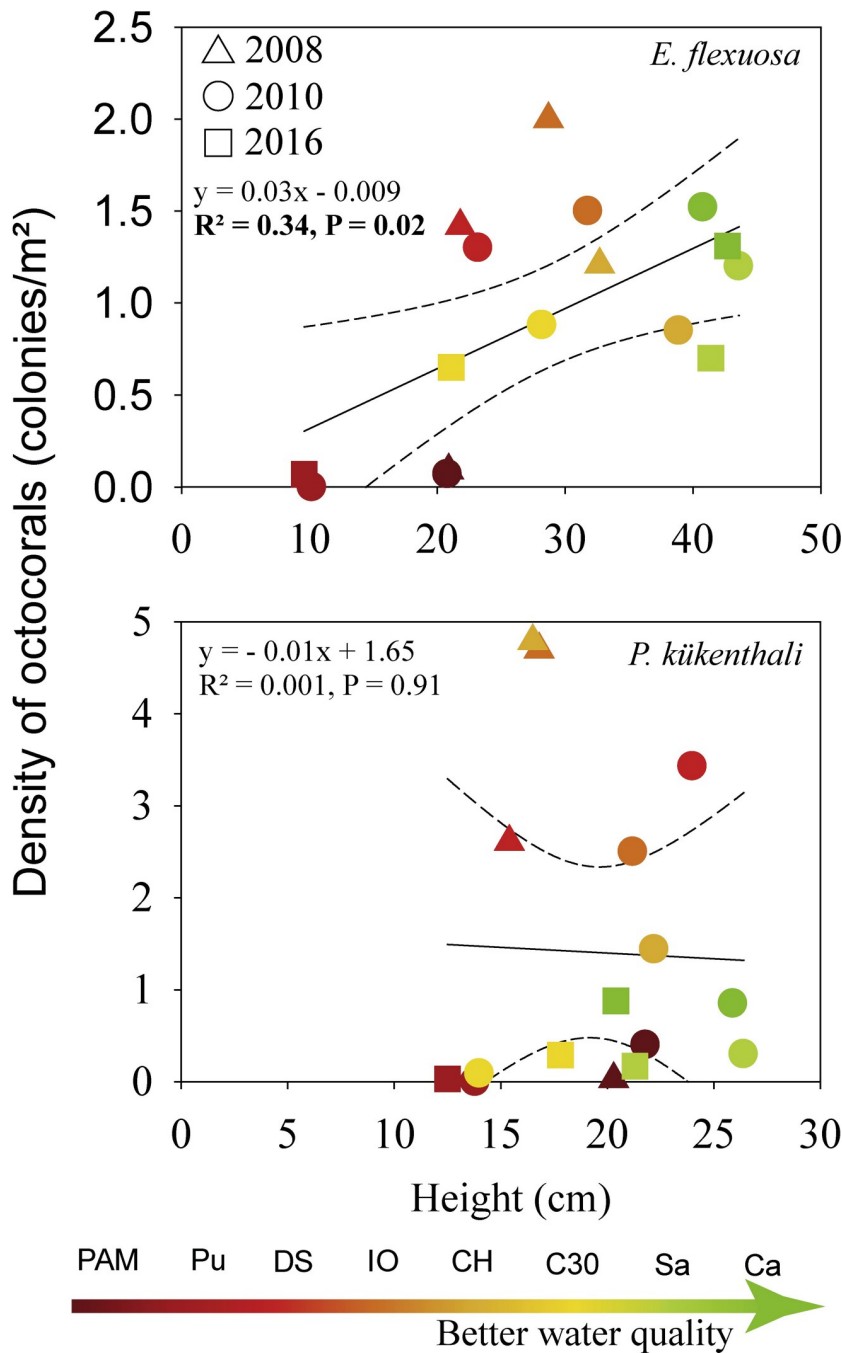

**Fig 8.** Linear regression between the density and height of *E. flexuosa* (top) and *P. kükenthali* (bottom) in the sampling years. The dotted lines represent the upper and lower 95% confidence intervals.

Therefore, this indicates that the decrease in the *E. flexuosa* height is not a morphological adaptation to the water quality gradient since its abundance is also affected. In the case of *P. kükenthali*, its abundance did not decrease as a function of variations in its height along the water quality gradient (Fig 8). This indicates that tolerance in the morphometry of *P. kükenthali* to water quality degradation may be one of the biological traits that contributes to explaining its abundance under these environmental conditions.

The vertical morphological strategy of octocorals minimizes the amount of resources committed to maintaining a place in the substrate and avoids competition with other benthic organisms for space as well as scour by sediments, shading by macroalgae and overgrowth by other benthic organisms [6, 16]. The tolerance of certain species in terms of density (e.g., *P. kükenthali*) [12] and vertical morphology to water quality degradation can contribute to the increase of octocoral abundance in Caribbean reefs [6, 7]. Some species of Caribbean octocorals have been classified as tolerant to organic pollution, while others are considered sensitive, based on their relative abundances in sites influenced by polluted river basins compared to reference sites. Species considered tolerant are *P. kükenthali*, *Pseudoplexaura flagellosa*, *Eunicea tourneforti*, *Eunicea calyculata forma calyculata*, *Eunicea calyculata forma coronata* [27], *Eunicea mammosa* [39] and *Pterogorgia citrina* [40]. Furthermore, the octocorals *E. tourneforti* and *Pseudoplexaura porosa* have shown tolerance to ammonium enrichment [15]. Sensitive species to organic pollution are *G. ventalina*, *Gorgonia mariae*, *Antillogorgia elisabethae*, *Antillogorgia rigida*, *A. americana* and *Muriceopsis flavida* [27, 39]. The tolerance of octocorals to organic pollution and their vertical growth, are life strategies that may contribute to the abundance of these species. In fact, the eight octocoral species tolerant to organic pollution and ammonium enrichment accounted for 56.5% of the total abundance of octocoral communities in sites influenced by polluted river basins on the Havana coast between 2008 and 2017. On the other hand, sensitive species contributed only 7.1% to the total abundance of octocoral communities under these environmental conditions [41]. Therefore, based on the results observed in *P. kükenthali* and *E. flexuosa*, we propose that other octocorals species tolerant in terms of density and vertical morphology to water quality degradation may experience increased abundances in scenarios of escalating anthropogenic pollution in Caribbean reefs.

Unlike the pattern observed with water quality, hydrodynamic stress appears to negatively affect the morphometry of *P. kükenthali*. The lowest means in height, diameter, number of terminal branches/colony, cover index and highest height/diameter ratio were detected at the sites with the highest hydrodynamic stress (e.g., Pu and C30, see HSI in S1 Table). *P. kükenthali* is considered a species that does not tolerate hydrodynamic stress and sediment abrasion conditions [33], factors that seem to control the distribution of this species in the reefs of the Havana coastline [12]. However, hydrodynamic stress does not seem to affect the morphometry of *E. flexuosa*, which is due to its tolerance to this factor [33, 42]. This shows that hydrodynamic stress also specifically influences the morphometry of *E. flexuosa* and *P. kükenthali*, as occurs in Mediterranean octocorals [30, 31]. For example, the height of *Eunicella singularis* was significantly greater at the site most exposed to wave in the North Aegean Sea, Greece [31], while the length of the terminal branches of *Eunicella cavolinii* was significantly greater in a wave-sheltered area in Corsica northwest, France [30].

The ecological implications of the effect of the water quality gradient on the *E. flexuosa* morphometry are the decrease in its abundance and the habitat loss for other organisms. *E. flexuosa* has been considered eurytopic because it is widely distributed among biotopes, depths (between 5–20 m) and reefs around Cuba (189 sites), except at sites under the influence of polluted river basins, where its abundance decreases [12, 41]. The reduction in *E. flexuosa* abundance and size under these environmental conditions can reduce the habitat for other organisms since this species tends to grow to taller heights than *P. kükenthali*. Although the height of *E. flexuosa* (30–41 cm) has been reported to be similar to that of *P. kükenthali* (39 cm) [1, 43], our results showed that *E. flexuosa* can grow up to 101 cm in height, while *P. kükenthali* only reached 61 cm, with the former species having greater cover (see Fig 2). Therefore, given the increase in anthropogenic pollution experienced by the Caribbean coastal reefs [44–46], our results suggest that may occur changes in octocoral communities toward more tolerant and smaller species that provide less habitat for different organisms. Thus, more

tolerant species such as *P. kükenthali* could provide habitat for various organisms in a scenario of increasing anthropogenic pollution.

## Chronic effect of water quality degradation on the height of *E. flexuosa* and *P. kükenthali*

The size distribution of octocorals is the result of long-term variations in environmental conditions such as water flow, light availability, temperature, nutrients, sedimentation, and water turbidity [13, 23, 47]. Our results show that the chronic effect of water quality degradation is lower height dispersion and higher percentages of colonies in the smaller size intervals in *E. flexuosa* in the reefs under the influence of permanent discharges from the Havana Bay and the Almendares and Quibú rivers. This finding coincides with the prevalence of small colonies and few tall colonies in the size distribution of *E. flexuosa* and *A. americana* on a reef with lower light availability and greater sedimentation in the U.S. Virgin Islands, Caribbean Sea [23]. In addition, the size distribution of other cnidarians, such as the hermatypic coral *Siderastrea siderea*, has also shown a prevalence of colonies with small sizes in polluted sites on the Havana coast [48, 49]. In contrast, the lower dispersion in height and the higher percentage of *P. kükenthali* colonies in the lower size intervals seem to be due to hydrodynamic stress (e.g., at Pu and C30, see S1 Table), since the water quality gradient did not explain the variability in the morphometry of this species. In fact, the size distribution of *P. kükenthali* in one of the polluted sites with less water visibility but with the least hydrodynamic stress (e.g., DS, see S1 Table) did not show significant differences with the sites with better water quality. This suggests that the water quality degradation and low visibility do not seem to affect the height of this species and that hydrodynamic stress is the factor that most influences its morphometry over time. This differential response in the size distribution of *E. flexuosa* and *P. kükenthali* along the water quality gradient has been observed in other octocoral species. For example, the symmetrical size distributions of the octocorals *Sinularia* sp. and *Sarcophyton* sp. show that the different size classes are well represented and may indicate a healthy population. However, the size distribution with negative skewness in *Lobophytum* sp. may indicate an eventual population decline with increased nutrients, sedimentation, and decreased water clarity in the Bolinao-Anda Reef Complex, Philippines [13].

The size distribution of octocorals is also a consequence of the differences in demographic processes such as recruitment, growth rate and mortality of the colonies [23, 50, 51]. The percentage of recruits of *E. flexuosa* at the sites with the highest organic pollution was < 2% (e.g., PAM, Pu and DS), and no recruits of *P. kükenthali* were detected at the sites with the highest hydrodynamic stress, with the recruits' percentage of both species being similar among all the sites (see Figs 4 and 5). This suggests that the trend in the size distribution of *E. flexuosa* and *P. kükenthali* under these environmental conditions does not seem to be related to an increase in recruitment. The growth rate in height of *E. flexuosa* and *P. kükenthali* was significantly higher at PAM than at Sa and Ca due to the greater contribution of organic matter, microorganisms, and nutrients [52]. Furthermore, the growth rate of *E. flexuosa* was higher in another of the polluted sites (e.g., DS), while for *P. kükenthali* it was similar between one of the polluted sites (e.g., DS) and reference sites (e.g., Sa and Ca) (S4 Table). These data show that the growth rate of both species does not decrease in coastal reefs under the influence of discharge from the Havana Bay and the Quibú River. Therefore, the highest percentages of colonies in the smallest size ranges and the very few colonies taller than 40 cm in height suggest a higher mortality of adult *E. flexuosa* colonies at the sites with the highest organic pollution and higher mortality of adult colonies of *P. kükenthali* at the sites with the highest hydrodynamic stress. This finding coincides with the results of Tsounis et al. [23], who found that the greater abundance of

colonies of *G. ventalina* between 30–40 cm in height was due to the low mortality of adult colonies in the U.S. Virgin Islands, Caribbean Sea.

The two-factor permutational analysis of variance showed that the water quality gradient more strongly explained the variability in the height of *E. flexuosa* than time. The lower heights of *E. flexuosa* in the most polluted sites during the 2008–2016 period appear to be due to the chronic effect of constant discharges from the Havana Bay and the Almendares and Quibú rivers. In fact, the linear regression model showed that the water quality degradation significantly explained the decrease in all the morphometric indicators of this species. This decrease in the height of *E. flexuosa* over time seems to be due to the chronic influence of pollution and not to morphological plasticity, as discussed in the previous section. Conversely, the similar effect of the water quality gradient and time on the height of *P. kükenthali* suggests that there does not seem to be a clear chronic effect of pollution on the height of this species as occurred in *E. flexuosa*. In fact, there were no significant differences in the height of *P. kükenthali* between one of the polluted sites (e.g., DS) and the sites with better water quality (e.g., Sa and Ca) over time. This coincided with the results of Grigg [17], who found that the height of *Muricea* sp. was greater at a site near an underwater sewage outfall in San Onofre, California. In addition, the lowest heights of *P. kükenthali* over time were detected only at sites under the chronic influence of hydrodynamic stress, a factor that significantly explained the decrease in all morphometric indicators in this species. This confirms the differentiated response of the octocorals morphometry to the water quality degradation and hydrodynamic stress over time, which is apparently related to their life strategies.

## Conclusions

The five morphometric indicators of *E. flexuosa* were affected by the water quality degradation, low visibility, and sediment accumulation on the reefs under the influence of polluted river basins. This suggests that there is no differentiated response between the morphometric indicators of this species along the water quality gradient. However, the water quality gradient did not explain the variability of the five morphometric indicators of *P. kükenthali*; hydrodynamic stress was the factor that most negatively affected its morphometry. The chronic effect of the water quality degradation over time (2008–2016) was the decrease in the height of *E. flexuosa*, and this was not due to its morphological plasticity since the abundance of this species decreased, which causes a reduction in habitat for other organisms. In the case of *P. kükenthali*, there was no clear evidence of the chronic effect of water quality degradation on its height. This finding indicates that there is a differentiated response in the morphometry of these octocoral species to water quality degradation, which is apparently related to the sensitivity of *E. flexuosa* and the tolerance of *P. kükenthali* to this factor. In fact, the tolerance in the morphometry of *P. kükenthali* to the chronic water quality degradation seems to be one of the biological traits that contributes to explaining its greater abundance under these environmental conditions. The vertical morphological strategy of octocorals together with the tolerance of certain species in terms of density and morphometry to increasing anthropogenic pollution may contribute to explaining the success of the group in Caribbean coastal reefs.

## Supporting information

**S1 Fig. Photos of the two species and of the measured morphometric indicators.** Photos of *Eunicea flexuosa* (A), *Plexaura kükenthali* (B), height from the holdfast to the colony apex (C), height from first branching to the colony apex (D), maximum diameter (E), and minimum diameter (F) of the colony from a top view. Photos credits: José Espinosa (C, D, E and F) and

Néstor Rey-Villiers (A and B).
(PDF)

**S2 Fig. Accumulated curves of the standard error versus the number of colonies for the height (H) of *Eunicea flexuosa* and *Plexaura kükenthali* at sites and years of sampling (2008, 2010 and 2016).**
(PDF)

**S3 Fig. Relative size-frequency (based on height) distributions of *E. flexuosa* between years at each of the sampling sites.** The statistics results of PERMDISP test are shown. N: sampling sizes. Codes of the sampling sites are presented in Fig 1.
(PDF)

**S4 Fig. Relative size-frequency (based on height) distributions of *P. kükenthali* between years at each of the sampling sites.** The statistics results of PERMDISP test are shown. N: sampling sizes. Codes of the sampling sites are presented in Fig 1.
(PDF)

**S1 Table. Mean (minimum-maximum) of microbiological, hydrochemical, and physical variables and stable nitrogen isotopes of octocorals at sampling sites (10 m of depth) over the 2008–2016 period.** Data of bottom-sediment accumulation (adimensional) were of 2011, and data of stable nitrogen isotopes of octocorals belong to 2016. FC: fecal coliform bacteria, TC: total coliform bacteria, FE: fecal streptococcal bacteria, HB: heterotrophic bacteria, SR: sulfate-reducing bacteria and DIN: dissolved inorganic nitrogen. Codes of sampling sites are presented in Fig 1. These data were published in Rey-Villiers et al. (2020; 2021a), except the hydrodynamic stress index.
(XLSX)

**S2 Table. Pearson´s correlation between the morphometric indicators of *E. flexuosa* and the microbiological, hydrochemical, and physical variables and stable nitrogen isotopes of *E. flexuosa*.** FC: fecal coliform bacteria, HB: heterotrophic bacteria, TC: total coliform bacteria, SR: sulfate-reducing bacteria, FE: fecal streptococcal bacteria, DIN: dissolved inorganic nitrogen, and HSI: hydrodynamic stress index. The correlations in black indicate P values < 0.05.
(PDF)

**S3 Table. Pearson´s correlation between the morphometric indicators of *P. kükenthali* and the microbiological, hydrochemical, and physical variables and stable nitrogen isotopes of *P. kükenthali*.** FC: fecal coliform bacteria, HB: heterotrophic bacteria, TC: total coliform bacteria, SR: sulfate-reducing bacteria, FE: fecal streptococcal bacteria, DIN: dissolved inorganic nitrogen, and HSI: hydrodynamic stress index. The correlations in black indicate P values < 0.05.
(PDF)

**S4 Table. Mean growth rate (cm/year) in height (95% CI) of *E. flexuosa* and *P. kükenthali* at sites influenced by discharges from polluted river basins and sites not impacted by those basins.**
(PDF)

**S5 Table. Raw data, mean and 95% CI of the morphometric indicators of *E. flexuosa* and *P. kükenthali* at each site and year of sampling.**
(XLSX)

## Acknowledgments

We sincerely thank Pedro Alcolado Prieto, Jorge Oliva, Johannes Acosta, and Sandy León for their valuable assistance in the fieldwork. We are also very grateful for the translation and editing provided to us by the International Chair for Coastal and Marine Studies in Cuba, Harte Research Institute for Gulf of Mexico Studies, Texas A&M University-Corpus Christi. We appreciate the insightful comments of two anonymous referees which improved the manuscript.

## Author Contributions

**Conceptualization:** Néstor Rey-Villiers, Alberto Sánchez, Patricia González-Díaz, Lorenzo Álvarez-Filip.

**Data curation:** Néstor Rey-Villiers.

**Formal analysis:** Néstor Rey-Villiers.

**Funding acquisition:** Néstor Rey-Villiers, Alberto Sánchez, Patricia González-Díaz.

**Investigation:** Néstor Rey-Villiers.

**Methodology:** Néstor Rey-Villiers.

**Project administration:** Alberto Sánchez, Patricia González-Díaz.

**Resources:** Néstor Rey-Villiers, Alberto Sánchez, Patricia González-Díaz.

**Supervision:** Alberto Sánchez, Patricia González-Díaz, Lorenzo Álvarez-Filip.

**Validation:** Alberto Sánchez, Patricia González-Díaz, Lorenzo Álvarez-Filip.

**Visualization:** Néstor Rey-Villiers, Lorenzo Álvarez-Filip.

**Writing – original draft:** Néstor Rey-Villiers.

**Writing – review & editing:** Alberto Sánchez, Patricia González-Díaz, Lorenzo Álvarez-Filip.

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
