## [Decision Letter · Decision Letter 0]

24 May 2023

PONE-D-23-00568Morphometric responses of two zooxanthellate octocorals along a water quality gradient in the Cuban northwestern coastPLOS ONE

Dear Dr. Rey Villiers,

Thank you for submitting your manuscript to PLOS ONE. After careful consideration, we feel that it has merit but does not fully meet PLOS ONE’s publication criteria as it currently stands. Therefore, we invite you to submit a revised version of the manuscript that addresses the points raised during the review process.

The reviewers agree that this is worthy of publication, but that there are some areas that need to be addressed before this can be accepted. Please see the reviewers comments below. Thank you!

We look forward to receiving your revised manuscript.

Kind regards,

Nikki Traylor-Knowles, Ph.D.

Academic Editor

PLOS ONE

3. We note that [Figure 1] in your submission contain [map/satellite] images which may be copyrighted. All PLOS content is published under the Creative Commons Attribution License (CC BY 4.0), which means that the manuscript, images, and Supporting Information files will be freely available online, and any third party is permitted to access, download, copy, distribute, and use these materials in any way, even commercially, with proper attribution. For these reasons, we cannot publish previously copyrighted maps or satellite images created using proprietary data, such as Google software (Google Maps, Street View, and Earth). For more information, see our copyright guidelines: http://journals.plos.org/plosone/s/licenses-and-copyright.

Natural Earth (public domain): http://www.naturalearthdata.com/.

Additional Editor Comments:

Dear Dr. Nèstor Rey Villiers,

I want to apologize for the length of time that it took for these reviews to come back to you. It has been challenging to find reviewers. Thank you for your patience.

Sincerely,

Nikki Traylor-Knowles

Reviewers' comments:

Reviewer's Responses to Questions

**Comments to the Author**

1. Is the manuscript technically sound, and do the data support the conclusions?

Reviewer #1: Yes

Reviewer #2: Partly

2. Has the statistical analysis been performed appropriately and rigorously? 

Reviewer #1: Yes

Reviewer #2: I Don't Know

3. Have the authors made all data underlying the findings in their manuscript fully available?

Reviewer #1: Yes

Reviewer #2: No

4. Is the manuscript presented in an intelligible fashion and written in standard English?

Reviewer #1: Yes

Reviewer #2: Yes

5. Review Comments to the Author

Reviewer #1: This is a novel manuscript that examines the morphology of two octocoral species in the north of Cuba. The corals occur across a gradient of anthropogenic influence, and the study shows interesting and convincing evidence of site/water quality related influences on the growth of the corals. I found the study design and analysis to be sound and the results interesting and insightful in terms of the influence of water quality.

The study builds off of a previous investigation of the study sites, including water quality work, which builds off of the previous Rey-Veilliers work (citations 12, 25). In the introduction, it would be helpful to set the stage of the previous work and then mention how this new work builds upon and extends that work. Since some of the data seem to be used in multiple publications (or else are related), it would be helpful to better explain these connections. For example, in lines 126-151 of the materials and methods, the authors present the overarching results of the water quality analysis, which is usually material that is presented in the Results. I understand that the authors are not actually showing that data here, but rather related data. Thus, the authors just need to make these differences clearer to the reader.

The manuscript was overall very well written and I appreciate the time that the non-native English speakers took to make the work presentable. I found a few sentences that were oddly worded, and offer suggestions below.

Line 37: Change to ‘Octocoral abundance is increasing on Caribbean reefs’

Line 40: change to ‘The aim of this study was to determine….’

Line 42: change to ‘on eight forereefs of northwest Cuba’

Line 43: change to ‘within a belt transect’

Lines 47/48: change to ‘hydrodynamic stress was the factor most negatively affecting P. kukenthali morphophometry’

Line 48: define hydrodynamic stress

Lines 48/49: The chronic effect of poor water quality over time resulted in more small sized colonies at the polluted site’

Line 53/54: change ‘water quality degradation’ to ‘poor water quality’

Lines 53-56: change to ‘This study suggests that poor water quality decreases the size and thus availability of octocoral habitat (e.g., E. flexuosa) while other tolerant species (e.g., P. kukenthali) are more tolerant of pollution.

Lines 116-117: change to ‘does the chronic effect of decreased water quality impact the height of both species over 9 years?’

Lines 118-120: Here it indicates that you will be describing the water quality measurements that you took for this study – this is part of the reason it is confusing as to what data was presented previously compared to what data is novel or set within a new context in this manuscript.

Reviewer #2: Review comments for “Morphometric responses of two zooxanthellate octocorals along a water quality gradient in the Cuban northwestern coast”

COMMENTS TO AUTHORS

Summary

This manuscript review examines the morphometric response of two octocoral species, Eunicea flexuosa and Plexaura kükenthali, in a water quality gradient within Caribbean reefs. The authors investigate the potential causes behind the increasing abundance of octocorals in this region, focusing on the species' vertical morphological plasticity and their ability to adapt to environmental gradients. The study was conducted between 2008 and 2016, encompassing eight fore reefs in the Cuban northwestern area. Various morphometric indicators were measured in colonies of both species along belt transects at each site. The findings reveal that E. flexuosa exhibited lower morphometric values (height, diameter, number of terminal branches/colony, cover index, and arborescent colonies) in areas with higher anthropogenic pollution. In contrast, the morphometry of P. kükenthali was primarily affected by hydrodynamic stress rather than water quality, although the authors’ methods for determination of hydrodynamic stress are unclear. Over the study period, water quality degradation led to a higher percentage of smaller-sized colonies of E. flexuosa, likely due to increased mortality. Similarly, the size distribution of P. kükenthali exhibited a similar trend but correlated with sites experiencing greater hydrodynamic stress. Overall, the study sheds light on the complex relationships between water quality, morphometric characteristics, and octocoral species dynamics in Caribbean reefs, and represents a valuable contribution that should be accepted following major revisions to provide context and clarity.

Major comments

1. Please expand the introduction and discussion to highlight the systematic relationship between the octocoral species studied here – how might these results extrapolate to other octocoral species or genera with comparable life histories and morphological traits? Other genera and their tolerances to various stressors are casually mentioned throughout the manuscript, but it is difficult for the reader to understand how these octocorals are related to one another and how different responses to water quality can impact their ecological success.

2. I would encourage the authors to include photographs of their study species, and a depiction of the morphometric characters that are being measured in the current study. This will help readers understand the differences in morphology that exist between the species. Representative photos showing the differences in morphology between E. flexuosa along the water gradient would also help illustrate the morphometric changes discussed in the manuscript.

3. Lastly, I politely recommend that the authors revise sentences for proper English grammar and improve phrasing for clarity and conciseness throughout the manuscript. There are many cases of extraneous prepositional phrases and inverted sentence structure that make it difficult to understand the main points of a paragraph. Also, the repetition of paragraph structure for results presented for E. flexuosa and P. kükenthali makes it hard to determine which species is the focus.

Minor comments

Abstract

Line 37: a grammatical correction is needed here: “The abundance of octocorals is increasing on Caribbean reefs...”

Line 40: correct to “The aim of this study was to determine...”

Line 47 – 48: consistent use of the correct tense is needed here: “..., however hydrodynamic stress was the factor that most negatively affected the morphometry of P. kükenthali.”

Introduction

Lines 60 – 61: The first sentence has awkward phrasing, please consider revising, for example, “Coral reef ecosystems are biodiversity hotspots that support high biological productivity and provide goods and services to society.”

Lines 61 – 63: Consider rephrasing to “Octocorals contribute substantially to supporting coral reef biodiversity and are the second most common group of sessile organisms on the reefs of the Caribbean and Indo-Pacific.”

Line 65: give the definition of “hermatypic” (reef-building) here in parentheses for unfamiliar readers, or use “scleractinian corals” for taxonomic clarity throughout the manuscript.

Lines 78 – 80: The evidence that shows certain octocorals’ morphologies are differentially susceptible to water quality should be discussed in more detail – which species/taxa exhibit these responses, and to which specific water quality parameters?

Line 95: What is the systematic relationship between P. kükenthali and E. flexuosa? Understanding the evolutionary relatedness of the study species will help the readers contextualize the results. Do these species have other differences in life history strategies?

Lines 108 – 110: Consider rephrasing to “Therefore, other morphometric indicators including the total length of the branches, height, diameter, height/diameter ratio, cover index and surface area are measured to determine octocoral size.” Also, two sentences starting with “Therefore” in a row is awkward and not appropriate.

Lines 116 – 117: rephrase poor wording, consider “Does the chronic effect of water quality degradation result in a decrease in height of both species over a 9 year (2008-2016 period)?”

Methods

Lines 131 – 132: rephrase for clarity, consider “The sampling sites were located on fore reefs approximately 120 to 700 m from the coast, at a depth of 10 m (Fig 1).”

Lines 171 – 174: Can you provide more information on how the HSI was determined? Which 11 octocoral species are used? If the HSI is not independent of P. kükenthali relative abundances, how can the morphometric response of P. kükenthali be driven by the hydrodynamic stress? This methodological approach and the consequences for the manuscript’s conclusions must be made clearer.

Lines 195 – 196: Please also report the mean and standard deviation of numbers of colonies per site.

Lines 219 – 221: Move the sentence on testing for normality earlier in the paragraph, it reads as an afterthought here.

Results

Lines 252 – 253: unclear, change to “similar variation as the colony diameter”, also was this lack of difference in variation confirmed with a statistical test?

Lines 295 – 297: Report which test was used to determine the percent variation in morphometric indicators explained by the water quality gradient, and clarify phrasing with the next sentence, consider “... E. flexuosa, indicating that water quality...”

Line 305: correct to “significant negative correlation” and cite a figure or table that contains these results.

Line 307: correct to “significant positive correlation” here and wherever this occurs (Lines 309, 310, 312, etc.)

Discussion

Lines 390 – 393: This sentence is too long and confusing, please separate the two discussion points or rephrase to state the conclusion first, followed by the lines of supporting evidence.

Lines 400 – 404: Also unclear, please correct for clarity and conciseness – what does “latter” refer to? Consider replacing use of “coincide” with a different phrase, for example “is similar to”.

Lines 412 – 418: Provide more specific evidence that suggests there are differences in tolerance of organic pollution between species. The linkages between the data, results, and conclusions here are tenuous and require more explanation.

Lines 422 – 426: These sentences are unclear, but it seems obvious that decreases in abundance would also result in smaller sizes of remaining colonies. Rephrase and clarify.

Lines 438 – 440: This sentence is not concise and needs to be edited to remove unnecessary prepositional phrases (“in the abundance”, “of the group”, etc.).

Lines 441 – 443: This sentence hangs on to the end of the paragraph and needs to be given more context or be removed – the genera Eunicea and Pseudoplexaura are not the focus of this study, and so the fact that these octocorals have developed tolerance mechanisms to heavy metals in the Red Sea must be put in context with the impacts of the water quality gradient in Cuba.

Lines 541 – 542 & Lines 548 – 549: Please explain the apparent contradiction in conclusions and clarify these sentences.

6. PLOS authors have the option to publish the peer review history of their article (what does this mean?). If published, this will include your full peer review and any attached files.

Reviewer #1: No

Reviewer #2: No

---

## [Author Response · Author response to Decision Letter 0]

19 Jun 2023

Reviewers´ comments:

Reviewer´s Responses to Questions

Comments to the Author

3. Have the authors made all data underlying the findings in their manuscript fully available?

Reviewer #1: Yes

Reviewer #2: No

Response to reviewer # 2: The supplementary material, Table Excel S5, has been updated. This Excel sheet displays the raw data, mean, and 95% confidence interval (CI) of the morphometric measurements for both species across the sampling sites and years. 

Response to reviewers

Response to reviewer #1

Reviewer: This is a novel manuscript that examines the morphology of two octocoral species in the north of Cuba. The corals occur across a gradient of anthropogenic influence, and the study shows interesting and convincing evidence of site/water quality related influences on the growth of the corals. I found the study design and analysis to be sound and the results interesting and insightful in terms of the influence of water quality.

Response: We thank the reviewer for considering the novelty of our manuscript.

Reviewer: The study builds off of a previous investigation of the study sites, including water quality work, which builds off of the previous Rey-Villiers work (citations 12, 25). In the introduction, it would be helpful to set the stage of the previous work and then mention how this new work builds upon and extends that work. Since some of the data seem to be used in multiple publications (or else are related), it would be helpful to better explain these connections. For example, in lines 126-151 of the materials and methods, the authors present the overarching results of the water quality analysis, which is usually material that is presented in the Results. I understand that the authors are not actually showing that data here, but rather related data. Thus, the authors just need to make these differences clearer to the reader.

Response: We agree with the reviewer. We have added three sentences to the end of the introduction (see lines 119-124) and two sentences in the material and methods section (see lines 142-145) to clarify this issue. 

Reviewer: Line 37: Change to ‘Octocoral abundance is increasing on Caribbean reefs’

Response: Done

Reviewer: Line 40: change to ‘The aim of this study was to determine….’ 

Response: Done 

Reviewer: Line 42: change to ‘on eight forereefs of northwest Cuba’

Response: Done 

Reviewer: Line 43: change to ‘within a belt transect’

Response: Done

Reviewer: Lines 47/48: change to ‘hydrodynamic stress was the factor most negatively affecting P. kukenthali morphophometry’ 

Response: Done

Reviewer: Line 48: define hydrodynamic stress

Response: It is the turbulent movement of water generated by waves (Alcolado, 1981; Hernández and Alcolado, 2007; Alcolado et al., 2008; Espinosa et al., 2010; Pérez-Angulo and de la Nuez, 2011; Kupfner and Hallock, 2020). In the materials and methods section, we expanded the description of how this indicator was obtained in response to this comment and to one from the second reviewer (see lines 176-180).

Reviewer: Lines 48/49: The chronic effect of poor water quality over time resulted in more small sized colonies at the polluted site’

Response: Done

Reviewer: Line 53/54: change ‘water quality degradation’ to ‘poor water quality’

Response: Done

Reviewer: Lines 53-56: change to ‘This study suggests that poor water quality decreases the size and thus availability of octocoral habitat (e.g., E. flexuosa) while other tolerant species (e.g., P. kukenthali) are more tolerant of pollution. 

Response: We modified the last sentence of the abstract based on the reviewer's recommendation (see lines 53-56). 

Reviewer: Lines 116-117: change to ‘does the chronic effect of decreased water quality impact the height of both species over 9 years?’

Response: Done

Reviewer: Lines 118-120: Here it indicates that you will be describing the water quality measurements that you took for this study – this is part of the reason it is confusing as to what data was presented previously compared to what data is novel or set within a new context in this manuscript. 

Response: No new data on water quality indicators were collected in this research. Instead, we used a gradient previously described by Rey-Villiers et al. (2020; 2021) to assess its influence on the morphometry of E. flexuosa and P. kükenthali. We have clarified these aspects by adding three sentences at the end of the introduction (see lines 119-124) and two sentences in the materials and methods section (see lines 142-145). 

Response to reviewer #2

Reviewer: This manuscript review examines the morphometric response of two octocoral species, Eunicea flexuosa and Plexaura kükenthali, in a water quality gradient within Caribbean reefs. The authors investigate the potential causes behind the increasing abundance of octocorals in this region, focusing on the species' vertical morphological plasticity and their ability to adapt to environmental gradients. The study was conducted between 2008 and 2016, encompassing eight fore reefs in the Cuban northwestern area. Various morphometric indicators were measured in colonies of both species along belt transects at each site. The findings reveal that E. flexuosa exhibited lower morphometric values (height, diameter, number of terminal branches/colony, cover index, and arborescent colonies) in areas with higher anthropogenic pollution. In contrast, the morphometry of P. kükenthali was primarily affected by hydrodynamic stress rather than water quality, although the authors’ methods for determination of hydrodynamic stress are unclear. Over the study period, water quality degradation led to a higher percentage of smaller-sized colonies of E. flexuosa, likely due to increased mortality. Similarly, the size distribution of P. kükenthali exhibited a similar trend but correlated with sites experiencing greater hydrodynamic stress. Overall, the study sheds light on the complex relationships between water quality, morphometric characteristics, and octocoral species dynamics in Caribbean reefs, and represents a valuable contribution that should be accepted following major revisions to provide context and clarity.

Response: We sincerely thank the reviewer for considering our study valuable. We have clarified the method for determining hydrodynamic stress on lines 176-180. 

Major comments

Reviewer: Please expand the introduction and discussion to highlight the systematic relationship between the octocoral species studied here – how might these results extrapolate to other octocoral species or genera with comparable life histories and morphological traits? Other genera and their tolerances to various stressors are casually mentioned throughout the manuscript, but it is difficult for the reader to understand how these octocorals are related to one another and how different responses to water quality can impact their ecological success.

Response: We understand the reviewer´s concern and have expanded our discussion by incorporating other octocoral species that share similar life strategies as those documented. This information allows us to extrapolate that other octocoral species tolerant to organic pollution and with vertical growth could increase their abundances in scenarios of heightened anthropogenic pollution in Caribbean reefs. These findings are discussed in detail on lines 450-467 of the discussion section. 

E. flexuosa and P. kükenthali are two closely related species of the Plexauridae family (see lines 96-97). However, we did not expand on the systematic relationship between both species, as this is unrelated to the objective of this research. Our focus is on the life history strategies of species. We selected E. flexuosa and P. kükenthali because they exhibit different life strategies and are present in all sampling sites.

We believe the referee’s suggestion rely on the likelihood that taxonomically related species are can also share life history strategies (and ecological traits). However, in octocorals there is evidence suggesting that closely related species do not always share environmental preferences. We provide some examples below. Overall, the evidence indicates that even species that are phylogenetically closely related show different life strategies. For these reasons, we do not consider the systematic relationships of E. flexuosa and P. kükenthali relevant in the context of this research, but rather their life strategies.

For example, there are species of octocorals within the same genus (e.g., Eunicea sp.) with a close phylogenetic relationship that show different bathymetric ranges and life strategies (Alcolado, 1981; Sánchez, 2009). Even different species of the genus Eunicea that form the same monophyletic group also present different bathymetric distribution and life strategies (Sánchez, 2009). For example: 

The octocorals E. flexuosa, Eunicea calyculata, Eunicea asperula, and Eunicea clavigera are species that are in the same monophyletic group (Sánchez, 2009) but have different bathymetric distributions and life strategies. 

The species E. flexuosa and E. calyculata inhabit reefs from 0.5-30 m depth with moderate to high water movement (Alcolado, 1981; Sánchez, 2009).

E. asperula and E. clavigera inhabit leeward reefs between 10-30 m and 2-55 m depth, respectively (Alcolado, 1981; Sánchez, 2009).

Two other taxonomically related species, Plexaura kükenthali and Plexaura homomalla, exhibit differences in their life history strategies, including reproductive time, mode of larval development, and symbiotic communities across multiple sampling sites in the Caribbean (Pelosi et al., 2022). 

Reviewer: I would encourage the authors to include photographs of their study species, and a depiction of the morphometric characters that are being measured in the current study. This will help readers understand the differences in morphology that exist between the species. Representative photos showing the differences in morphology between E. flexuosa along the water gradient would also help illustrate the morphometric changes discussed in the manuscript.

Response: We have made modifications to Figure S1 by adding a photo for each species and including four photos of the measured morphometric indicators in the colonies of both species. Descriptions of the measured morphometric indicators can be found in materials and methods section. We agreed with the reviewer´s suggestion that including photos of the morphometric differences of both species along the water quality gradient would help illustrate the discussed changes. Unfortunately, we do not have photos of both species along the gradient. 

Reviewer: Lastly, I politely recommend that the authors revise sentences for proper English grammar and improve phrasing for clarity and conciseness throughout the manuscript. There are many cases of extraneous prepositional phrases and inverted sentence structure that make it difficult to understand the main points of a paragraph. Also, the repetition of paragraph structure for results presented for E. flexuosa and P. kükenthali makes it hard to determine which species is the focus.

Response: Done. The manuscript was edited for proper English language by a highly qualified native English-speaking editor at American Journal Experts (AJE). Therefore, it was also edited by a native English speaker. 

Minor comments

Abstract

Line 37: a grammatical correction is needed here: “The abundance of octocorals is increasing on Caribbean reefs...”

Response: Done

Line 40: correct to “The aim of this study was to determine...”

Response: Done

Line 47–48: consistent use of the correct tense is needed here: “..., however hydrodynamic stress was the factor that most negatively affected the morphometry of P. kükenthali.” 

Response: Done

Introduction

Lines 60–61: The first sentence has awkward phrasing, please consider revising, for example, “Coral reef ecosystems are biodiversity hotspots that support high biological productivity and provide goods and services to society.”

Response: Done

Lines 61–63: Consider rephrasing to “Octocorals contribute substantially to supporting coral reef biodiversity and are the second most common group of sessile organisms on the reefs of the Caribbean and Indo-Pacific.” 

Response: Done

Line 65: give the definition of “hermatypic” (reef-building) here in parentheses for unfamiliar readers, or use “scleractinian corals” for taxonomic clarity throughout the manuscript.

Response: Done 

Lines 78–80: The evidence that shows certain octocorals’ morphologies are differentially susceptible to water quality should be discussed in more detail – which species/taxa exhibit these responses, and to which specific water quality parameters?

Response: We agree with the reviewer´s suggestion. We have already discussed the findings of the cited works (12, 13, 17, 18) in the introduction and discussion of the manuscript. The evidence related to the fact that certain species show a specific morphometric response to water quality (specifying water quality parameters) (citations 17 and 18) has already been highlighted in the article (see lines 94-96 and 557-559). The water quality parameters in both previous works refer to organic pollution and wastewater discharges from an underwater outfall (these are the terms used in both publications). On the other hand, we have added to the discussion the discoveries made by Lala et al. (2021) related to the specific morphometric response of certain octocoral species to water quality changes (specifying the parameters). These findings are discussed in detail on lines 517-523 of the discussion section. 

Line 95: What is the systematic relationship between P. kükenthali and E. flexuosa? Understanding the evolutionary relatedness of the study species will help the readers contextualize the results. Do these species have other differences in life history strategies?

Response: They are two closely related species of the Plexauridae family (see lines 96-97). In addition, these questions essentially mirror the observations made in Reviewer 2's initial comment. Therefore, we already answered these questions earlier in that first comment. The beginning of the answer to those questions was: we understand the reviewer´s concern and have expanded our discussion by incorporating other octocoral species that share similar life strategies as those documented. 

Lines 108–110: Consider rephrasing to “Therefore, other morphometric indicators including the total length of the branches, height, diameter, height/diameter ratio, cover index and surface area are measured to determine octocoral size.” Also, two sentences starting with “Therefore” in a row is awkward and not appropriate.

Response: Done

Lines 116–117: rephrase poor wording, consider “Does the chronic effect of water quality degradation result in a decrease in height of both species over a 9 year (2008-2016 period)?” 

Response: Done

Methods

Lines 131–132: rephrase for clarity, consider “The sampling sites were located on fore reefs approximately 120 to 700 m from the coast, at a depth of 10 m (Fig 1).”

Response: Done

Lines 171–174: Can you provide more information on how the HSI was determined? Which 11 octocoral species are used? If the HSI is not independent of P. kükenthali relative abundances, how can the morphometric response of P. kükenthali be driven by the hydrodynamic stress? This methodological approach and the consequences for the manuscript’s conclusions must be made clearer. 

Response: The HIS is independent of the relative abundance of Plexaura kükenthali. This species is considered sensitive to hydrodynamic stress and is not included in the index calculations. This index is a proxy for hydrodynamic stress and is calculated from the sum of the relative abundance of 11 octocoral species considered tolerant to this factor (Alcolado, 1981). This author sampled octocoral communities at 193 coral reef sites around Cuba. Sites were characterized with respect to wind direction (i.e., windward, and leeward sites), depth (between 1-30 m), and shelf direction relative to prevailing winds. With the above information and the variability in the relative abundance of octocorals, Alcolado (1981) determined the species tolerant and sensitive to hydrodynamic stress. The tolerant species are, Eunicea mammosa, Eunicea tourneforti, Eunicea calyculata, Plexaurella dichotoma, Muricea muricata, Gorgonia flabellum, Gorgonia ventalina, Eunicea flexuosa, Pterogorgia anceps, Pterogorgia citrina and Pterogorgia guadalupensis. This index has already been used in other research in marine ecology such as Hernández and Alcolado (2007), Alcolado et al. (2008), Espinosa et al. (2010), Pérez-Angulo and de la Nuez (2011) and Kupfner and Hallock (2020). 

Lines 195–196: Please also report the mean and standard deviation of numbers of colonies per site.

Response: In those lines, we indicate the range of the number of colonies of both species measured in the study area. While we can calculate the average number of colonies and the standard deviation of both species for the study area, we believe it is more important to provide the specific number of colonies measured for each site. This information can be found in the raw data presented in Table S5. 

Lines 219–221: Move the sentence on testing for normality earlier in the paragraph, it reads as an afterthought here.

Response: Done 

Results

Lines 252–253: unclear, change to “similar variation as the colony diameter”, also was this lack of difference in variation confirmed with a statistical test? 

Response: All the morphometric indicators of E. flexuosa exhibited significant variation along the water quality gradient, as confirmed by the analysis of variance (PERMANOVA). This indicates that there was no differential response in the morphometric indicators, as they all showed a similar trend. In other words, there was no specific response of the E. flexuosa morphometric indicators along the gradient.

Lines 295–297: Report which test was used to determine the percent variation in morphometric indicators explained by the water quality gradient, and clarify phrasing with the next sentence, consider “... E. flexuosa, indicating that water quality...” 

Response: The test used to determine the percent variation in morphometric indicators explained by the water quality gradient was R2 (see Fig. 3). We clarified that sentence (see line 298) according to the reviewer's suggestion.

Line 305: correct to “significant negative correlation” and cite a figure or table that contains these results. 

Response: Done 

Line 307: correct to “significant positive correlation” here and wherever this occurs (Lines 309, 310, 312, etc.)

Response: Done

Discussion

Lines 390–393: This sentence is too long and confusing, please separate the two discussion points or rephrase to state the conclusion first, followed by the lines of supporting evidence.

Response: Done

Lines 400–404: Also unclear, please correct for clarity and conciseness – what does “latter” refer to? Consider replacing use of “coincide” with a different phrase, for example “is similar to”.

Response: Done. We clarify the sentence (see line 405). 

Lines 412–418: Provide more specific evidence that suggests there are differences in tolerance of organic pollution between species. The linkages between the data, results, and conclusions here are tenuous and require more explanation.

Response: Specific evidence regarding the differences in tolerance to organic pollution between E. flexuosa and P. kükenthali has already been published in the work titled, "Spatio temporal variation in octocoral assemblages along a water quality gradient in the northwestern region of Cuba" (Rey-Villiers et al., 2020). To provide additional information and clarify this topic, we have included the following paragraph (see lines 419-424). 

"This is attributed to the significantly lower density of E. flexuosa colonies (colonies/m²) at sites influenced by polluted river basin discharges compared to reference sites. Furthermore, the density of E. flexuosa showed a significant negative correlation with several microbiological and physical variables [12]. As for P. kükenthali, its density increased at sites affected by polluted river basin discharges and did not show significant differences between the reefs near Havana Bay (PAM) and the reference sites (Sa and Ca) [12]". 

Lines 422–426: These sentences are unclear, but it seems obvious that decreases in abundance would also result in smaller sizes of remaining colonies. Rephrase and clarify. 

Response: The sentence has been rephrased and clarified in lines 430-435. It is important to note that the decrease in the morphometric indicators of E. flexuosa is not attributed to morphological plasticity, as the species has demonstrated in other environmental gradients (Kim et al., 2004; Prada et al., 2008). In this case, it is a consequence of organic pollution, as it significantly affects the abundance of the specie. On the other hand, this is not the case for P. kükenthali. The height of P. kükenthali ranges approximately between 15-25 cm across several sites with similar abundance (see Fig. 8). In fact, there are sites where the height of this species decreases, but its abundance remains unaffected (see Fig. 8).

Lines 438–440: This sentence is not concise and needs to be edited to remove unnecessary prepositional phrases (“in the abundance”, “of the group”, etc.).

Response: This sentence was edited (see lines 448-450). 

Lines 441–443: This sentence hangs on to the end of the paragraph and needs to be given more context or be removed – the genera Eunicea and Pseudoplexaura are not the focus of this study, and so the fact that these octocorals have developed tolerance mechanisms to heavy metals in the Red Sea must be put in context with the impacts of the water quality gradient in Cuba.

Response: We agree with the reviewer´s suggestion, so we have modified this sentence (see lines 450-467). 

Lines 541–542 & Lines 548–549: Please explain the apparent contradiction in conclusions and clarify these sentences.

Response: There is no contradiction in the conclusions presented in those lines. In lines 541-542 (currently lines 569-570), we refer to the fact that all the morphometric indicators of E. flexuosa showed the same trend, without any specific response of these indicators along the water quality gradient (see lines 398-400). On the other hand, in lines 548-549 (currently lines 576-579), we refer to the species-specific response on morphometry of E. flexuosa and P. kükenthali along the gradient. Specifically, the morphometric indicators in E. flexuosa had a different response compared to those of P. kükenthali.

---

## [Decision Letter · Decision Letter 1]

7 Aug 2023

Morphometric responses of two zooxanthellate octocorals along a water quality gradient in the Cuban northwestern coast

PONE-D-23-00568R1

Dear Dr. Rey Villiers,

We’re pleased to inform you that your manuscript has been judged scientifically suitable for publication and will be formally accepted for publication once it meets all outstanding technical requirements.

Kind regards,

Nikki Traylor-Knowles, Ph.D.

Academic Editor

PLOS ONE

Additional Editor Comments (optional):

Reviewers' comments:

Reviewer's Responses to Questions

**Comments to the Author**

1. If the authors have adequately addressed your comments raised in a previous round of review and you feel that this manuscript is now acceptable for publication, you may indicate that here to bypass the “Comments to the Author” section, enter your conflict of interest statement in the “Confidential to Editor” section, and submit your "Accept" recommendation.

Reviewer #2: All comments have been addressed

2. Is the manuscript technically sound, and do the data support the conclusions?

Reviewer #2: Yes

3. Has the statistical analysis been performed appropriately and rigorously? 

Reviewer #2: Yes

4. Have the authors made all data underlying the findings in their manuscript fully available?

Reviewer #2: Yes

5. Is the manuscript presented in an intelligible fashion and written in standard English?

Reviewer #2: Yes

6. Review Comments to the Author

Reviewer #2: I am impressed with the thorough response to the previous comments, and recommend the manuscript for publication.

The introduction, methods, results and discussion sections are much improved, and benefit from a clearer contextualization of this work in relation to previous studies. I have also noted that you addressed some minor language and grammatical issues, which have improved the overall readability of the manuscript. Thank you for your diligence in improving the phrasing and logical flow of the manuscript's paragraphs.

The included images of the study species in the supplemental figure 1 are much appreciated, and help connect the reader to the differing geometry of the species under investigation. In addition, thank you for providing additional information about the different life history strategies of these species, and how within-genera evolutionary relationships may not offer clear explanations for octocorals' ecological niches, but rather specific life history strategies (such as height, morphology) are more important.

Considering the quality of your work and the efforts you have put into addressing the reviewers' feedback, I am pleased to accept your manuscript for publication.

7. PLOS authors have the option to publish the peer review history of their article (what does this mean?). If published, this will include your full peer review and any attached files.

Reviewer #2: No

---

## [Editor Report · Acceptance letter]

10 Aug 2023

PONE-D-23-00568R1 

Morphometric responses of two zooxanthellate octocorals along a water quality gradient in the Cuban northwestern coast 

Dear Dr. Rey-Villiers:

I'm pleased to inform you that your manuscript has been deemed suitable for publication in PLOS ONE. Congratulations! Your manuscript is now with our production department. 

Kind regards, 

on behalf of

Dr. Nikki Traylor-Knowles 

Academic Editor

PLOS ONE